# Progerin accelerates atherosclerosis by inducing endoplasmic reticulum stress in vascular smooth muscle cells

Magda R Hamczyk[1,2,3], Ricardo Villa-Bellosta[1,4], Víctor Quesada[3,5], Pilar Gonzalo[1], Sandra Vidak[6], Rosa M Nevado[1], María J Andrés-Manzano[1,2], Tom Misteli[6], Carlos López-Otín[3,5,*] [iD] & Vicente Andrés[1,2,**] [iD]

## Abstract

Hutchinson–Gilford progeria syndrome (HGPS) is a rare genetic disorder caused by progerin, a mutant lamin A variant. HGPS patients display accelerated aging and die prematurely, typically from atherosclerosis complications. Recently, we demonstrated that progerin-driven vascular smooth muscle cell (VSMC) loss accelerates atherosclerosis leading to premature death in apolipoprotein E-deficient mice. However, the molecular mechanism underlying this process remains unknown. Using a transcriptomic approach, we identify here endoplasmic reticulum stress (ER) and the unfolded protein responses as drivers of VSMC death in two mouse models of HGPS exhibiting ubiquitous and VSMC-specific progerin expression. This stress pathway was also activated in HGPS patient-derived cells. Targeting ER stress response with a chemical chaperone delayed medial VSMC loss and inhibited atherosclerosis in both progeria models, and extended lifespan in the VSMC-specific model. Our results identify a mechanism underlying cardiovascular disease in HGPS that could be targeted in patients. Moreover, these findings may help to understand other vascular diseases associated with VSMC death, and provide insight into aging-dependent vascular damage related to accumulation of unprocessed toxic forms of lamin A.

**Keywords** aging; endoplasmic reticulum stress; progeria; unfolded protein response; vascular smooth muscle cell

**Subject Categories** Ageing; Genetics, Gene Therapy & Genetic Disease; Vascular Biology & Angiogenesis

See also: **ED Pasquale & G Condorelli** (April 2019)

## Introduction

Hutchinson–Gilford progeria syndrome (HGPS) is a devastating disease with an estimated prevalence of 1 in 20 million people (www.progeriaresearch.org). Affected children appear normal at birth but show early onset of aging-associated symptoms, including alopecia, reduced subcutaneous fat, osteoporosis, joint stiffness, and dermal abnormalities (Hennekam, 2006; Gordon *et al*, 2007; Merideth *et al*, 2008). The most important clinical manifestation of the disease is atherosclerosis, which causes death from myocardial infarction or stroke at an average age of 14.6 years (Ullrich & Gordon, 2015). Progerin also provokes cardiac abnormalities (Merideth *et al*, 2008; Rivera-Torres *et al*, 2016; Prakash *et al*, 2018) and defects in heart valves (Nair *et al*, 2004; Merideth *et al*, 2008; Olive *et al*, 2010; Hanumanthappa *et al*, 2011) and blood vessels, including vascular smooth muscle cell (VSMC) loss, adventitial thickening, calcification, and extracellular matrix deposition (Stehbens *et al*, 1999, 2001; Olive *et al*, 2010).

"Classic" HGPS is caused by a point mutation in the *LMNA* gene (c.1824C>T;p.G608G), which activates a cryptic splice site in exon 11, leading to deletion of 150 nucleotides in the mRNA (De Sandre-Giovannoli *et al*, 2003; Eriksson *et al*, 2003). Consequent loss of 50 amino acids near the C terminus of the precursor prelamin A protein affects its post-translational modifications, resulting in the production of a permanently farnesylated mutant protein called progerin. Progerin alters many of the cellular functions normally regulated by lamin A, causing abnormal localization of nuclear envelope proteins, impaired chromatin organization, DNA damage and genome instability, mitochondrial dysfunction, oxidative stress, altered gene transcription and signal transduction, among others (Gordon *et al*, 2014b; Vidak & Foisner, 2016; Dorado & Andres, 2017). However, only a few studies have investigated the molecular alterations caused by progerin accumulation in VSMCs (Zhang *et al*, 2011, 2014; Villa-Bellosta *et al*, 2013; Hamczyk *et al*, 2018a), and

1  Centro Nacional de Investigaciones Cardiovasculares Carlos III (CNIC), Madrid, Spain
2  Centro de Investigación Biomédica en Red de Enfermedades Cardiovasculares (CIBERCV), Spain
3  Departamento de Bioquímica y Biología Molecular, Instituto Universitario de Oncología (IUOPA), Universidad de Oviedo, Oviedo, Spain
4  Fundación Instituto de Investigación Sanitaria Fundación Jiménez Díaz (FIIS-FJD), Madrid, Spain
5  Centro de Investigación Biomédica en Red de Cáncer (CIBERONC), Spain
6  Cell Biology of Genomes Group, National Cancer Institute, NIH, Bethesda, MD, USA
   *Corresponding author. Tel: +34 985104202; E-mail: clo@uniovi.es
   **Corresponding author. Tel: +34 914531200; E-mail: vandres@cnic.es

none analyzed it in the context of atherosclerosis, the main death-causing symptom of HGPS, due to lack of adequate animal models. Recently, we have generated an HGPS-like mouse model of ubiquitous progerin expression that reproduces the main features of human HGPS, including VSMC loss in the aortic media, adventitial thickening, accelerated atherosclerosis, and shortened lifespan (Hamczyk *et al*, 2018b). Moreover, we have shown that limiting progerin expression to VSMCs is sufficient to cause the loss of these cells in the aorta and accelerate atherosclerosis and death, demonstrating a key role of VSMCs in the pathogenesis of HGPS (Hamczyk *et al*, 2018b). The aim of this study was to identify molecular mechanisms underlying progerin-induced VSMC loss, the principal driver of premature atherosclerosis and death in HGPS.

## Results

To identify pathways underlying progerin-induced VSMC death, we conducted a transcriptomic analysis of progerin-expressing aortas. To this end, we used two mouse models of HGPS, with progerin expressed either ubiquitously ($Apoe^{-/-}Lmna^{G609G/G609G}$) or restricted to VSMCs ($Apoe^{-/-}Lmna^{LCS/LCS}SM22\alpha Cre$), which fully recapitulate the vascular phenotype observed in HGPS patients (Olive *et al*, 2010; Hamczyk *et al*, 2018b). To identify drivers of disease rather than secondary changes, we sought to collect arteries before the onset of evident disease. Since both substantial atherosclerosis and overt aortic structure alterations are found in normal chow-fed 16-week-old mice, but absent in normal chow-fed 8-week-old mice of both genotypes (Hamczyk *et al*, 2018b), we collected aortas at 8 weeks of age. To specifically detect molecular alterations in VSMC, which appear to be a major progerin target, we digested aortas with collagenase to separate fibroblast-containing adventitia from the VSMC-rich media (Fig 1A). We collected four pooled samples per genotype ($Apoe^{-/-}Lmna^{G609G/G609G}$ and $Apoe^{-/-}Lmna^{LCS/LCS}SM22\alpha Cre$ mice and their corresponding littermate controls $Apoe^{-/-}Lmna^{+/+}$ and $Apoe^{-/-}Lmna^{LCS/LCS}$, respectively), which showed the expected progerin and lamin A expression as assessed by PCR (Fig 1B).

Differential expression analysis of RNA sequencing data revealed 776 significantly altered genes in the ubiquitous progeroid model and 931 altered genes in the VSMC-specific model (Fig 1C, Datasets EV1 and EV2). Of these differentially regulated genes, 240 were common to both comparisons (Fig 1C, Dataset EV3) and exhibited high correlation ($R^2 \approx 0.8$, Fig 1D). Analysis of the two controls revealed 176 genes differentially expressed between $Apoe^{-/-}Lmna^{LCS/LCS}$ aorta (expressing lamin C only) and $Apoe^{-/-}Lmna^{+/+}$ aorta (expressing wild-type lamin A/C; Dataset EV4). However, there was barely any overlap between the gene sets affected by progerin production in $Apoe^{-/-}Lmna^{G609G/G609G}$ and $Apoe^{-/-}Lmna^{LCS/LCS}SM22\alpha Cre$ mice and those influenced by the lack of lamin A in $Apoe^{-/-}Lmna^{LCS/LCS}$ mice (Fig 1C). Likewise, we found no overlap in the main pathways affected by progerin expression (Fig EV1A and B) and lack of lamin A (Fig EV1C).

Comparison analysis identified four pathways that were significantly altered in medial aorta in both the ubiquitous and the VSMC-specific progerin-expressing models: fibrosis, nuclear factor erythroid 2-like 2 (NRF2)-mediated oxidative stress, endoplasmic reticulum (ER) stress response, and unfolded protein response (UPR; Fig 1E,

*Canonical pathways*). We also examined the predicted activation status of upstream regulators based on the expression of their target genes. This analysis revealed that the most differentially regulated factors belong to the ER stress response and ER stress-related UPR. These factors include X-box-binding protein 1 (XBP1), activating transcription factor 4 (ATF4; also known as cyclic AMP-dependent transcription factor ATF-4), eukaryotic translation initiation factor 2-alpha kinase 3 (EIF2AK3; also known as protein kinase RNA-like ER kinase; PERK), and DNA damage-inducible transcript 3 protein (DDIT3; also known as C/EBP-homologous protein, CHOP; Fig 1E, *Upstream regulators,* genes in each ER stress-related upstream regulator network are shown in Fig EV2A–F). Further overrepresentation test for gene ontology (GO) cellular compartment of 240 genes shared between ubiquitous and VSMC-specific progeroid models showed higher than 10-fold enrichment for sarcoplasmic reticulum (GO:0016529) and sarcoplasm (GO:0016528; Table EV1).

To validate the RNA sequencing results, we performed quantitative real-time PCR on selected ER stress response and UPR genes that were significantly upregulated in aortic media from 8-week-old mice in both progeria models (Fig 2A; see also schemes in Appendix Fig S1 showing genes within ER stress/UPR pathway significantly altered in each model). This analysis confirmed progerin-induced upregulation of *Calr*, *Ddit3*, *Dnajb9*, *Hspa5*, *Hsp90b1*, and *Pdia4* in VSMC-rich aortic media in both models (Fig 2B and C). We next used quantitative real-time PCR to analyze immortalized HGPS patient-derived cells. A wide range of ER stress-related genes, such as *HSP90B1*, *HSPA5*, *CALR,* and *DNAJC3*, and the *bona fide* UPR genes *DDIT3*, *ATF4*, *EIF2AK3*, *ERN1*, *PPP1R15A*, and the spliced form of *XBP1* were upregulated in HGPS patient cells (Fig 2D). We also assessed whether progerin activates the ER stress response and the UPR in other organs of our progeroid mouse models. Consistent with the ubiquity of progerin expression in $Apoe^{-/-}Lmna^{G609G/G609G}$ mice, induction of ER stress response and the UPR was noted in some organs of these animals, with kidney being the most affected organ and liver the least (Fig 2E). As anticipated, no activation of this stress pathway was detected in kidney, liver, spleen, or heart from $Apoe^{-/-}Lmna^{LCS/LCS}SM22\alpha Cre$ mice, consistent with the VSMC-specificity of the model (Fig 2F). The state of ER stress in VSMCs in the aorta of both progeroid mouse models was further confirmed by immunostaining against ER chaperone binding-immunoglobulin protein (BiP, also known as 78 kDa glucose-regulated protein, GRP78; product of *Hspa5* gene), calreticulin (product of *Calr* gene), and protein disulfide isomerase (PDI; product of *P4hb* gene; Fig 3A and B). Quantitative analysis revealed higher level of these proteins in the aorta of both progeroid mouse models compared with their respective controls (Fig 3C and D).

Our RNA sequencing results strongly suggested that progerin-induced ER stress response and UPR might underlie VSMC death and enhance atherosclerosis in the ubiquitous and VSMC-specific progeria mice. We therefore examined the potential benefits of tauroursodeoxycholic acid (TUDCA), a chemical chaperone that augments the capacity of cells to sustain ER stress and protects from apoptosis (Xie *et al*, 2002; Rivard *et al*, 2007; Uppala *et al*, 2017). One-week TUDCA treatment of 8-week-old $Apoe^{-/-}Lmna^{LCS/LCS}SM22\alpha Cre$ mice slightly increased in medial aorta the mRNA levels of *Pdia4* (coding for a protein disulfide isomerase) and *Hsp90b1* (coding for a chaperone; Fig EV3), suggesting mild enhancement of protein folding capacity. Moreover, it markedly

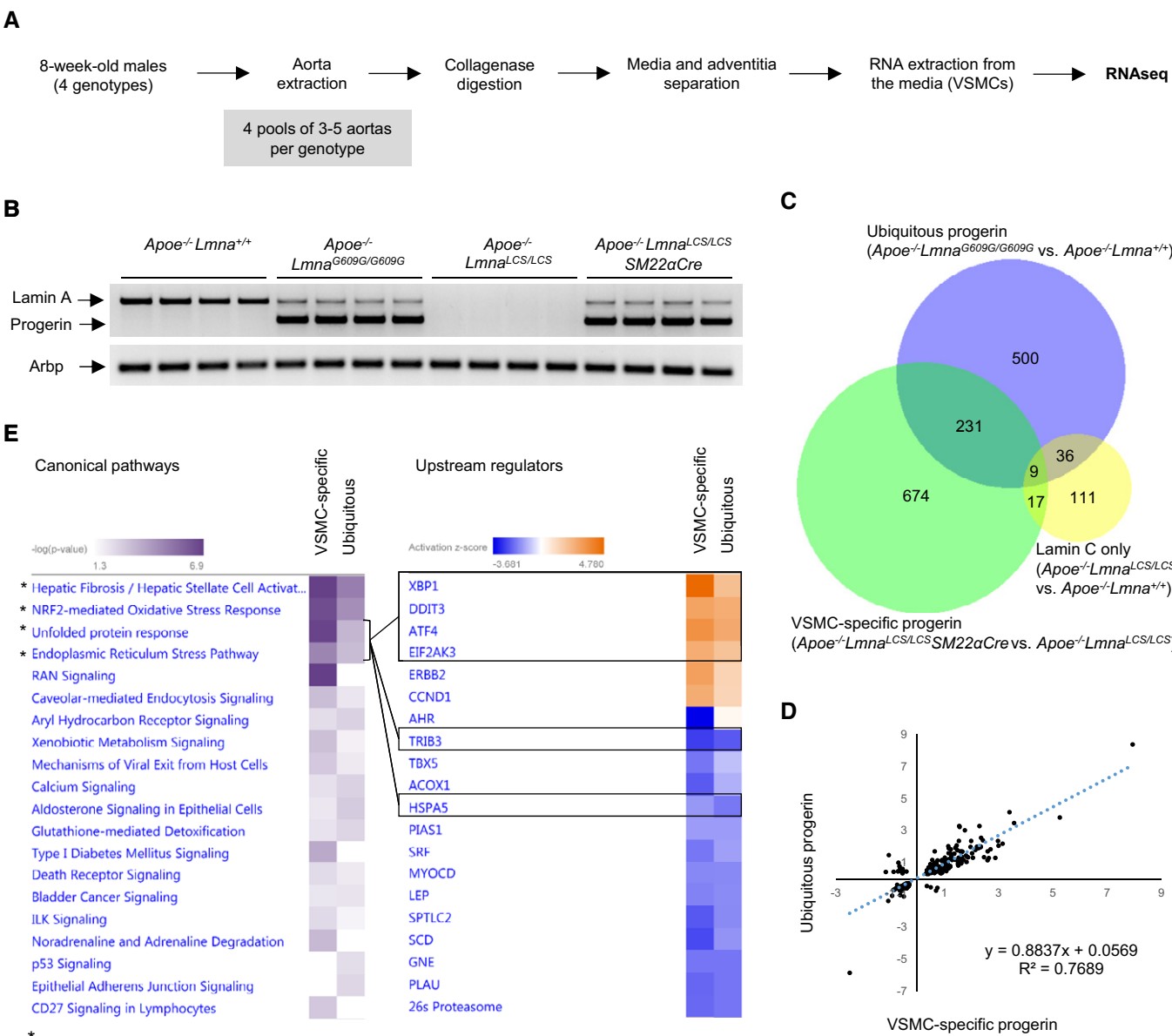

**Figure 1. Progerin expression in vascular smooth muscle cell (VSMC)-rich aortic media activates endoplasmic reticulum (ER) stress and unfolded protein response (UPR).**

A    Sample preparation for RNA sequencing (RNAseq).

B    PCR confirmation of proper expression of lamin A and progerin mRNA in pooled medial aortas used for RNAseq. *Arbp* was used as endogenous control.

C    Bioinformatic analysis detected 776 differentially expressed genes in medial aortas from *Apoe$^{−/−}$Lmna$^{G609G/G609G}$* mice with ubiquitous progerin expression compared with *Apoe$^{−/−}$Lmna$^{+/+}$* control mice expressing wild-type lamin A/C and 931 genes in medial aortas from *Apoe$^{−/−}$Lmna$^{LCS/LCS}$SM22αCre* mice with VSMC-specific progerin expression compared with *Apoe$^{−/−}$Lmna$^{LCS/LCS}$* control mice expressing lamin C only. There were 176 genes differentially expressed between the two control groups. The Venn diagram shows the overlap between sets of deferentially expressed genes identified in each of the three comparisons (*n* = 4 pooled medial aortas for each genotype).

D    Correlation between base-2 logarithms of fold change calculated for the 240 genes shared between the comparisons "ubiquitous progerin versus wild-type lamin A/C" and "VSMC-specific progerin versus lamin C only".

E    RNAseq results were analyzed using Ingenuity Pathway Analysis: (*left*) canonical pathway heatmap, showing processes affected by progerin expression in VSMC-rich medial aortas. Asterisk (*) indicates pathways which are significantly changed in both comparisons after applying the Benjamini–Hochberg correction for multiple testing; (*right*) upstream regulator heatmap, showing predicted activation states of transcriptional regulators (black boxes indicate key molecules involved in ER stress and UPR regulation). Genes in each upstream regulator network are shown in the Fig EV2.

Source data are available online for this figure.

decreased the expression of *Ddit3* gene coding for DDIT3 (Fig EV3), a pro-apoptotic transcription factor of the UPR machinery, indicating that TUDCA helps to resist ER stress-induced death in VSMCs.

After confirming that prolonged TUDCA administration did not trigger any deleterious side effects (Appendix Fig S2), we evaluated its effectiveness in ameliorating vascular disease in high-fat diet-fed ubiquitous and VSMC-specific progeroid mouse models. TUDCA

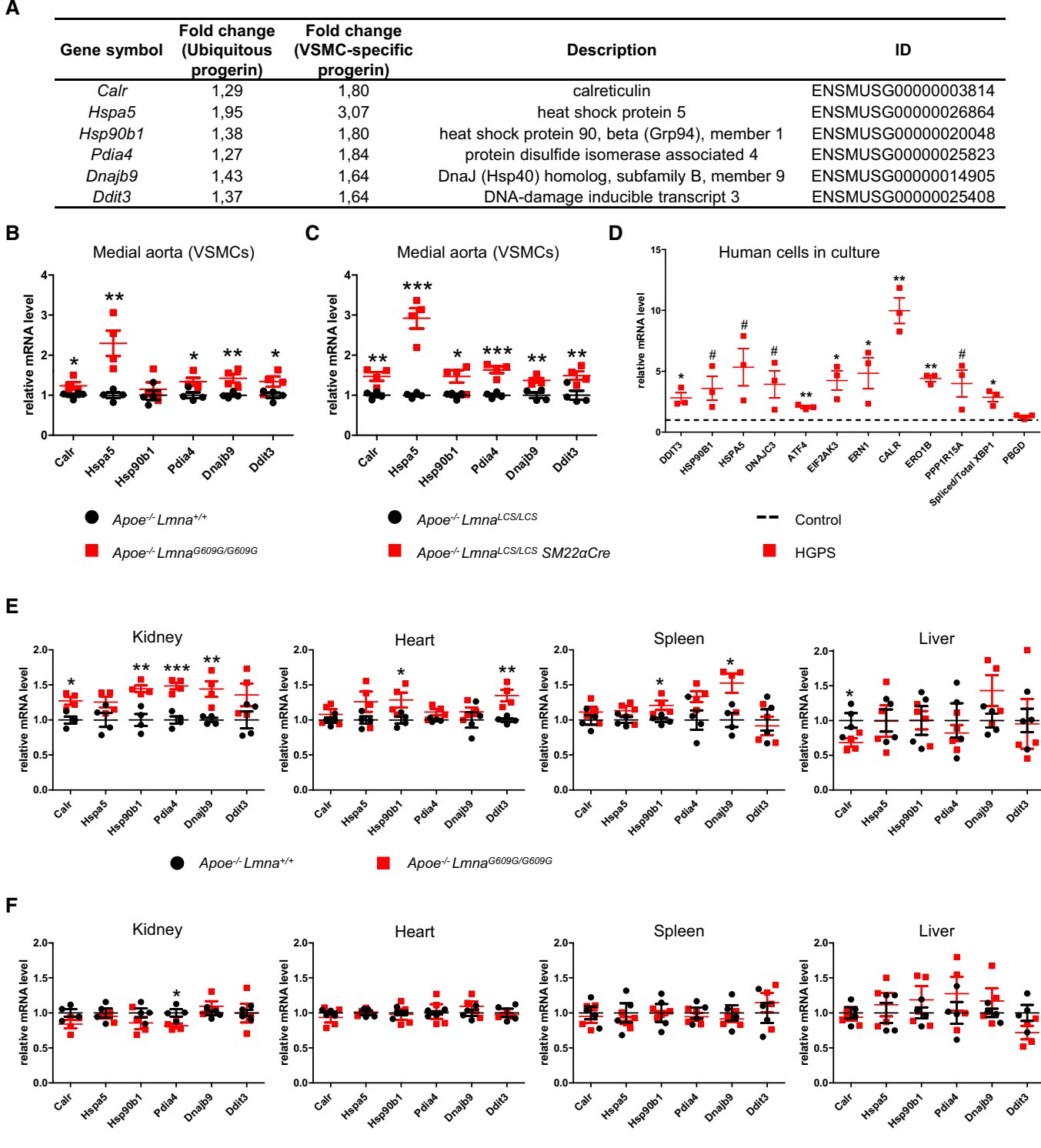

**Figure 2.**

**Figure 2.** Endoplasmic reticulum (ER) stress and unfolded protein response (UPR) activation in progerin-expressing medial aortas and in Hutchinson–Gilford progeria syndrome (HGPS) patient-derived cells.

A Six ER stress/UPR pathway genes selected for quantitative real-time PCR validation from among those detected as differentially expressed in RNAseq in both the ubiquitous and the vascular smooth muscle cell (VSMC)-specific models.

B, C mRNA levels of the selected genes in medial aortas obtained from 8-week-old $Apoe^{-/-}Lmna^{G609G/G609G}$ mice (B) and $Apoe^{-/-}Lmna^{LCS/LCS}SM22\alpha Cre$ mice (C) and their corresponding controls. *Hprt* and *Gusb* were used for normalization (n = 4 pools of medial aortas for each genotype).

D Quantification of mRNA levels of several ER stress and UPR genes in patient-derived immortalized human HGPS fibroblasts relative to healthy control. *GAPDH* was used for normalization and *PBGD* served as a negative control (n = 3 cultures for each group).

E, F mRNA levels of the selected genes in organs from 8-week-old $Apoe^{-/-}Lmna^{G609G/G609G}$ mice (E) and $Apoe^{-/-}Lmna^{LCS/LCS}SM22\alpha Cre$ mice (F) and their corresponding controls. *Hprt* and *Gusb* were used for normalization (n = 4 mice for each genotype).

Data information: In (B–F), data are presented as mean ± SEM. Statistical differences were analyzed by one-tailed unpaired *t*-test in (B, C), one-tailed one-sample *t*-test in (D), and by two-tailed unpaired *t*-test in (E, F). $^{\#}P < 0.061$, $^{*}P < 0.05$, $^{**}P < 0.01$, $^{***}P < 0.001$. The exact *P*-values are shown in Appendix Table S1.

treatment alleviated aortic VSMC loss (Fig 4A and B), adventitial thickening (Fig 4C and D), and inhibited atherosclerosis (Fig 4E and F) in both $Apoe^{-/-}Lmna^{G609G/G609G}$ and $Apoe^{-/-}Lmna^{LCS/LCS}SM22\alpha Cre$ mice. Atheromas of TUDCA-treated progeroid animals showed reduced necrotic core size and increased VSMC content (Table EV2), indicating an amelioration of the vulnerable plaque phenotype reported previously in these mice (Hamczyk *et al*, 2018b). We next assessed the effect of a sustained TUDCA treatment on survival in normal chow-fed progeroid mice. TUDCA prolonged the median lifespan of $Apoe^{-/-}Lmna^{LCS/LCS}SM22\alpha Cre$ mice by 38% (median survival: 64.15 weeks in TUDCA-treated versus 46.45 weeks in untreated mice), without significantly affecting the survival of $Apoe^{-/-}Lmna^{G609G/G609G}$ mice (Fig 4G).

## Discussion

Given the importance of VSMCs in progerin-driven cardiovascular disease (Stehbens *et al*, 2001; Olive *et al*, 2010; Hamczyk *et al*, 2018b), we sought to identify mechanisms underlying VSMC death and enhanced atherosclerosis in HGPS, which seem to be independent of elevated cholesterol levels in the blood (Gordon *et al*, 2005; Hamczyk *et al*, 2018b). Previous *in vitro* and *in vivo* studies identified numerous pathways potentially contributing to HGPS (Strandgren *et al*, 2017). Most of them are mutually dependent, making it difficult to distinguish primary from secondary mechanisms. Our RNAseq analysis of pre-disease aortic medial cells from the ubiquitous and VSMC-specific progeria models identified for the first time the ER stress response and the related UPR as potential driver mechanisms of VSMC death and the subsequent atherosclerosis in progeria. ER stress activation in other tissues of progeroid mice was variable. This variability might be at least in part attributed to different levels of lamin A (and thus progerin) expression associated with differences in tissue stiffness, with lower expression in softer tissues (Swift *et al*, 2013; Swift & Discher, 2014). Moreover, distinct organs and tissues may have different thresholds for tolerating misfolded protein load and therefore be more prone or resistant to ER stress.

Under ER stress conditions, cells activate UPR to restore homeostasis, but death occurs when stress cannot be resolved. Due to constant crosstalk between stress pathways, ER stress can lead to activation of other stress responses. For example, it can sensitize cells to DNA damage-induced apoptosis (Mlynarczyk & Fahraeus, 2014), providing a possible link between our findings and previous studies showing increased DNA damage in HGPS cells (Liu *et al*,

2013). In accordance with recent studies in HGPS fibroblasts that identified the NRF2 pathway as a major progerin target (Kubben *et al*, 2016), our RNAseq analysis detected NRF2-mediated oxidative stress response as one of the main pathways affected by progerin expression in VSMCs. Oxidative stress induced by NRF2 is connected with ER stress through the protein kinase EIF2AK3, which phosphorylates the transcription factor NRF2 in response to ER stress (Cullinan *et al*, 2003). Moreover, ER stress is associated with inflammation, autophagy, and mitochondrial dysfunction (Senft & Ronai, 2015), processes affected in progeria (Marino *et al*, 2008; Osorio *et al*, 2012; Rivera-Torres *et al*, 2013). Our results identify an upstream mechanism of great importance in VSMCs, which may link various stress pathways previously described in progeria. Future studies are warranted to assess whether progerin induces ER stress directly or indirectly, and to identify the exact mechanism of ER stress/UPR-related VSMC death triggered by progerin.

Our work also demonstrates that treatment with TUDCA, previously proven to efficiently alleviate ER stress in experimental diabetes (Ozcan *et al*, 2006), aortic valve calcification (Cai *et al*, 2013), and cardiac disease (Rivard *et al*, 2007; Rani *et al*, 2017), markedly ameliorated vascular pathology in the fat-fed mouse models with ubiquitous and VSMC-specific progerin expression ($Apoe^{-/-}Lmna^{G609G/G609G}$ and $Apoe^{-/-}Lmna^{LCS/LCS}SM22\alpha Cre$ mice, respectively). However, TUDCA did not prolong lifespan in $Apoe^{-/-}Lmna^{G609G/G609G}$ mice with ubiquitous progerin expression. This is in accordance with previous observations that, unlike HGPS patients, these mutant mice apparently die from atherosclerosis-independent causes, possibly arrhythmias, starvation, and cachexia (Hamczyk *et al*, 2018b; Kreienkamp *et al*, 2018). In contrast to $Apoe^{-/-}Lmna^{G609G/G609G}$ mice, $Apoe^{-/-}Lmna^{LCS/LCS}SM22\alpha Cre$ mice most likely die from atherosclerosis-related causes (Hamczyk *et al*, 2018b). In this setting, we have shown here that prevention of atherosclerosis by TUDCA was also associated with a significant increase of lifespan in $Apoe^{-/-}Lmna^{LCS/LCS}SM22\alpha Cre$ mice.

At the molecular level, 1-week *in vivo* TUDCA treatment caused a slight increase in the expression of some genes related to protein folding and modification in progerin-expressing medial aortas. Importantly, TUDCA diminished the mRNA levels of pro-apoptotic *Ddit3*, consistent with the known cytoprotective properties of this compound (Xie *et al*, 2002; Rivard *et al*, 2007; Gavin *et al*, 2016; Uppala *et al*, 2017). Overall, these results suggest that TUDCA stimulates the pro-survival UPR and attenuates the pro-apoptotic UPR; however, the exact molecular mechanism of action of TUDCA in progerin-expressing cells remains to be explored in further details.

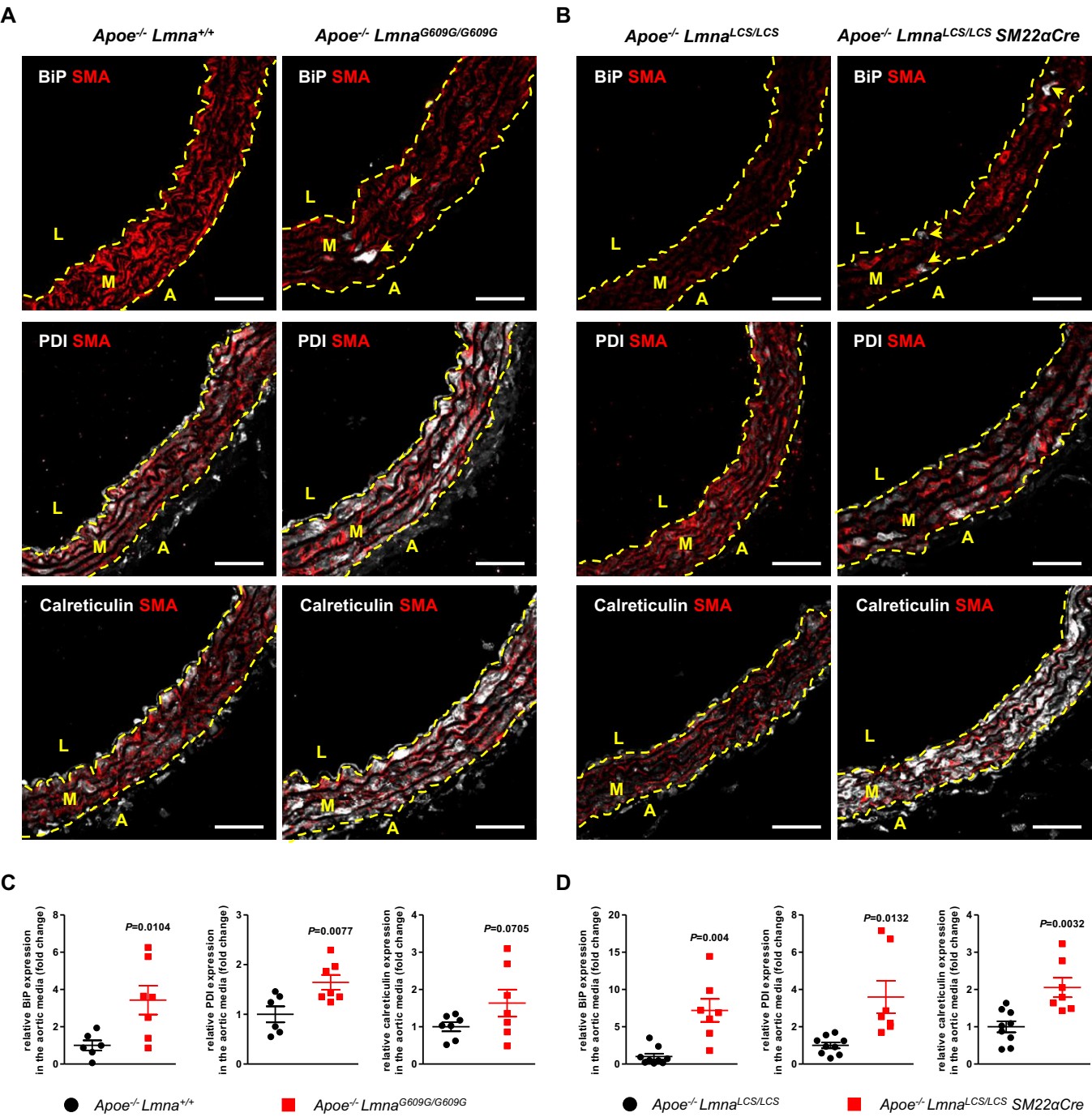

**Figure 3. Activation of endoplasmic reticulum stress in medial aortas of *Apoe*^−/−^*Lmna*^G609G/G609G^ and *Apoe*^−/−^*Lmna*^LCS/LCS^*SM22αCre* mice.**

A, B  Representative immunofluorescence images of aortas from 8-week-old *Apoe*^−/−^*Lmna*^G609G/G609G^ (A) and *Apoe*^−/−^*Lmna*^LCS/LCS^*SM22αCre* (B) mice and their corresponding controls stained with anti-α-smooth muscle actin (SMA) antibody (red), endoplasmic reticulum chaperone binding-immunoglobulin protein (BiP, white, *upper panel*), protein disulfide isomerase (PDI, white, *middle panel*), and calreticulin (white, *bottom panel*). Arrowheads indicate BiP-positive cells. Scale bar: 50 μm. L, lumen; M, media; A, adventitia.

C, D  Graphs show quantification of BiP, PDI, and calreticulin protein expression in medial aortas from 8-week-old *Apoe*^−/−^*Lmna*^G609G/G609G^ (C) and *Apoe*^−/−^*Lmna*^LCS/LCS^*SM22αCre* (D) mice relative to control mice (*n* = 6–9 mice for each genotype; aortic regions analyzed: aortic arch).

Data information: In (C, D), data are mean ± SEM. Statistical differences were analyzed by one-tailed unpaired *t*-test with Welch's correction.

The chemical chaperone TUDCA, a low-abundance bile acid in humans (Bentayeb *et al*, 2008), has been successfully used to treat cholestatic liver disease in humans (Crosignani *et al*, 1996; Larghi *et al*, 1997). Various studies reported no adverse effects of TUDCA treatment (daily doses from 500 to 1,750 mg per person) in obese, liver-transplanted, and primary biliary cirrhosis patients (Setchell

et al, 1996; Angelico et al, 1999; Invernizzi et al, 1999; Kars et al, 2010). In general, chemical chaperones show exceptional in vivo safety, and some of them have been approved for clinical use by the U.S. Food and Drug Administration, for example, 4-phenylbutyrate for urea cycle disorders (Maestri et al, 1996; Burlina et al, 2001) and ursodeoxycholic acid for primary biliary cirrhosis (Heathcote et al, 1994; Lindor et al, 1994; Poupon et al, 1999). Importantly, chemical chaperones have been used successfully in children

(Maestri et al, 1996; Burlina et al, 2001). Since the existing therapies for HGPS show only moderate therapeutic clinical benefits (Gordon et al, 2012, 2014a, 2016), our work suggests that the potential of chemical chaperone treatment to ameliorate atherosclerosis and associated cardiovascular events should be explored in children with HGPS. The use of a combination of drugs targeting different pathways may be an effective therapeutic strategy until approaches directly targeting progerin production become available in humans.

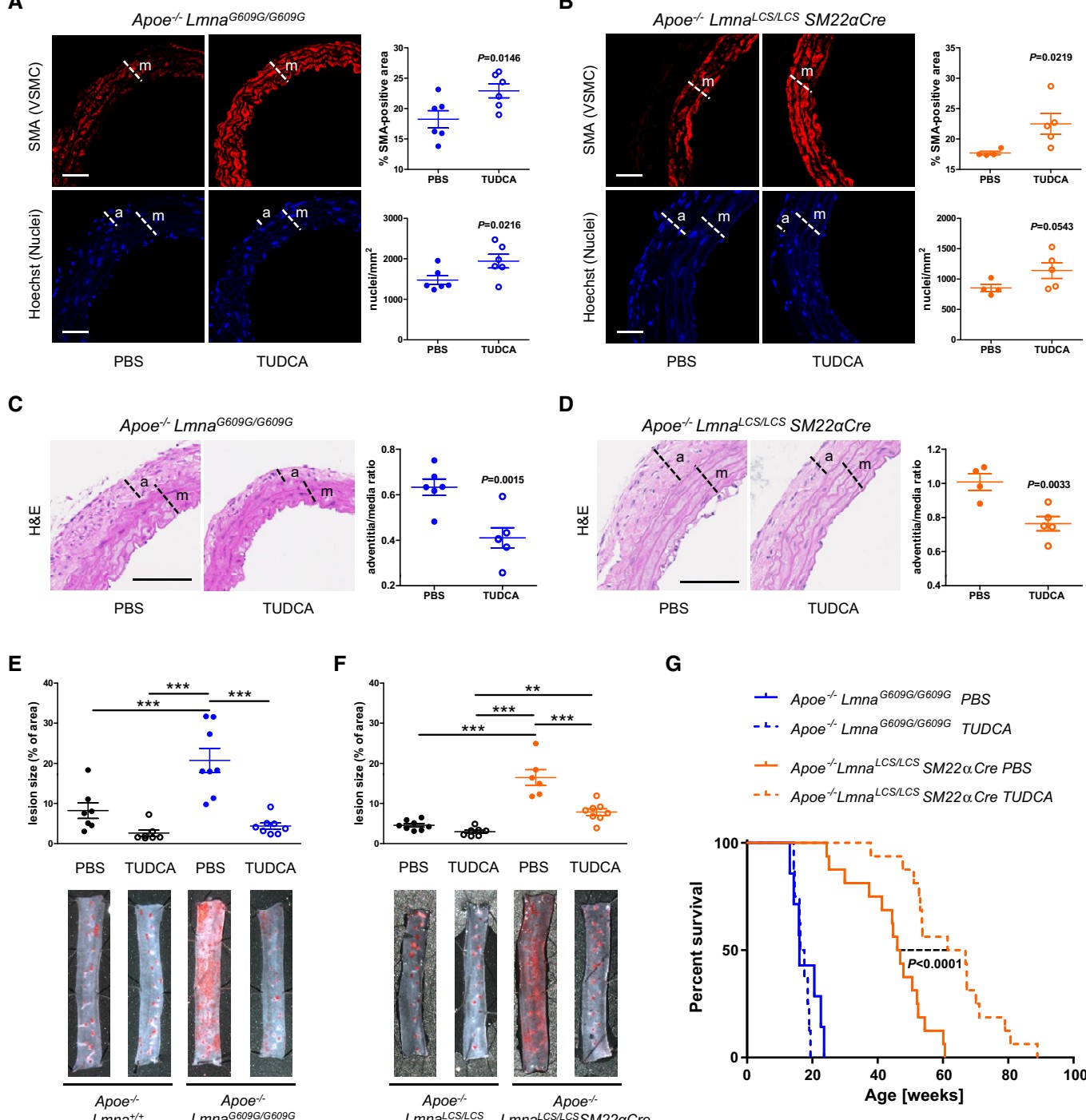

**Figure 4.**

**Figure 4.  Tauroursodeoxycholic acid (TUDCA) treatment alleviates vascular phenotype in progeria mouse models and extends lifespan in $Apoe^{-/-}$ $Lmna^{LCS/LCS}SM22\alpha Cre$ mice.**

Mice were injected 3 times a week with TUDCA or phosphate-buffered saline (PBS), starting at 6 weeks of age for $Apoe^{-/-}Lmna^{+/+}$ and $Apoe^{-/-}Lmna^{G609G/G609G}$ mice and at 8 weeks of age for $Apoe^{-/-}Lmna^{LCS/LCS}$ and $Apoe^{-/-}Lmna^{LCS/LCS}SM22\alpha Cre$ mice. In (A–F), mice were fed a high-fat diet for 8 weeks starting at 8 weeks of age and sacrificed at 16 weeks of age. In (G), mice were fed normal chow.

A, B   Representative immunofluorescence images of aortas stained with anti-α-smooth muscle actin (SMA) antibody (red) and Hoechst 33342 (blue). Graphs show quantification of vascular smooth muscle cell (VSMC) content in the media as either % of SMA-positive area (*top*) or nucleus count (*bottom*) ($n = 6$ mice for each group in A; $n = 4$–5 mice for each group in B; aortic regions analyzed: aortic arch and thoracic aorta). Scale bar: 50 μm. m, media; a, adventitia.

C, D   Representative histology sections of hematoxylin & eosin (H&E)-stained aortas. Graphs show quantification of adventitia-to-media thickness ratio ($n = 6$ mice for each group in C; $n = 4$–5 mice for each group in D; aortic regions analyzed: aortic arch and thoracic aorta). Scale bar: 100 μm. m, media; a, adventitia.

E, F   Representative images of thoracic aortas stained with Oil Red O and quantification of atherosclerosis burden in TUDCA-treated and untreated (PBS) fat-fed mice of the indicated genotypes ($n = 7$–8 mice for each group in E; $n = 6$–8 mice for each group in F).

G   Kaplan–Meier survival curves of TUDCA-treated and untreated (PBS) mice of the indicated genotypes ($n = 7$–8 $Apoe^{-/-}Lmna^{G609G/G609G}$ mice for each group; $n = 16$ $Apoe^{-/-}Lmna^{LCS/LCS}SM22\alpha Cre$ mice for each group); $P < 0.0001$ for TUDCA-treated vs untreated $Apoe^{-/-}Lmna^{LCS/LCS}SM22\alpha Cre$ mice (median survival: 64.15 vs 46.45 weeks, respectively).

Data information: Data in (A–F), are mean ± SEM. Statistical differences were analyzed by one-tailed unpaired $t$-test in (A–D), one-way ANOVA with Tukey's *post hoc* test in (E, F), and by log-rank test in (G). **$P < 0.01$, ***$P < 0.001$.

Source data are available online for this figure.

## Materials and Methods

### Study approval

Animal experimental procedures followed the EU Directive 2010/63EU and Recommendation 2007/526/EC, enforced in Spanish law under Real Decreto 53/2013. Animal protocols were approved by the local ethics committees and the Animal Protection Area of the Comunidad Autónoma de Madrid (PROEX167/16).

### Mice

All mice were housed in specific pathogen-free facility in individually ventilated cages (maximum of 5 animals per cage) with 12-h light/12-h dark cycle at a temperature of 22 ± 2°C, 50% of relative humidity (range of 45–60%). Mice had *ad libitum* access to water and food (5K67, LabDiet and D184, SAFE). Experimental mice used in this study were C57BL/6J males with ubiquitous ($Apoe^{-/-}Lmna^{G609G/G609G}$) and VSMC-specific ($Apoe^{-/-}Lmna^{LCS/LCS}SM22\alpha Cre$) progerin expression and their littermate controls with lamin A/C ($Apoe^{-/-}Lmna^{+/+}$) and lamin C only ($Apoe^{-/-}Lmna^{LCS/LCS}$) expression, respectively (Osorio *et al*, 2011; Hamczyk *et al*, 2018b). $Lmna^{LCS/LCS}$ mice used for the generation of VSMC-specific progeroid model show slightly increased weight and survival compared to wild-type controls (Lopez-Mejia *et al*, 2014). The $Lmna^{LCS}$ (Lamin C-STOP) allele consists of a modified *Lmna* gene in which a neomycin resistance gene flanked with two *loxP* sequences was introduced after exon 10 to abolish lamin A production. Additionally, an HGPS-equivalent point mutation (c.1827C>T; p.G609G) was inserted in exon 11. Excision of the neomycin resistance cassette in the presence of Cre recombinase enables progerin production (as well as lamin C and some residual lamin A). The specificity of $Apoe^{-/-}Lmna^{LCS/LCS}SM22\alpha Cre$ mouse model, which expresses Cre (and hence progerin) under a $SM22\alpha$ (*Tagln*, trans-gelin) promoter, was assessed previously in the aorta by immuno-histochemistry and in the heart, spleen, kidney, and liver by PCR (Hamczyk *et al*, 2018b). In the current manuscript, we examined by immunofluorescence the presence of progerin in embryonic tissues and in adult kidney and intestine of $Apoe^{-/-}Lmna^{LCS/LCS}SM22\alpha Cre$ mice (Appendix Figs S3–S8). At 11.5 days post coitum (dpc),

$Apoe^{-/-}Lmna^{LCS/LCS}SM22\alpha Cre$ embryo had undetectable progerin expression in the aorta (Appendix Fig S3A), intestine (Appendix Fig S4A) and kidney (Appendix Fig S5A), and very low level of expression was observed in some cells of the heart (Appendix Fig S6A). At 15.5 dpc, progerin expression in $Apoe^{-/-}Lmna^{LCS/LCS}SM22\alpha Cre$ embryos was detected principally in the aorta (Appendix Fig S3B) and, to a lesser extent, in the heart (predominantly evident in the valves, some expression was also observed in the ventricle wall, Appendix Fig S6B). Progerin expression in the intestine of $Apoe^{-/-}Lmna^{LCS/LCS}SM22\alpha Cre$ embryos at 15.5 dpc was limited to a few cells in the muscularis, but was undetectable in the mucosa (Appendix Fig S4B). Expression of progerin in the kidney of $Apoe^{-/-}Lmna^{LCS/LCS}SM22\alpha Cre$ embryos at 15.5 dpc was undetectable or extremely low (Appendix Fig S5B). Kidneys of adult $Apoe^{-/-}Lmna^{LCS/LCS}SM22\alpha Cre$ mice showed progerin expression predominantly restricted to arteries and arterioles, with low expression of the mutant protein in sparse cells outside the vessel wall (Appendix Fig S7). Progerin expression in the intestine of adult $Apoe^{-/-}Lmna^{LCS/LCS}SM22\alpha Cre$ mice was detected principally in the muscularis externa (Appendix Fig S8A and C), but smooth muscle cell loss was not observed in any of the regions analyzed in both the circular and the longitudinal muscle (Appendix Fig S8B and D). Sample size for animal studies was estimated based on our previous experience and mouse availability.

### Embryos

$Apoe^{-/-}Lmna^{LCS/LCS}SM22\alpha Cre$ and $Apoe^{-/-}Lmna^{LCS/LCS}$ mouse embryos were collected at 11.5 and 15.5 dpc. Day 0 of gestation was identified by the detection of the vaginal plug in the morning after overnight mating. Embryos were microdissected under a surgical stereoscopic microscope, and exsanguinated in phosphate-buffered saline (PBS) and heparin.

### Sample collection and preparation for RNA sequencing

Eight-week-old mice ($Apoe^{-/-}Lmna^{G609G/G609G}$, $Apoe^{-/-}Lmna^{+/+}$, $Apoe^{-/-}Lmna^{LCS/LCS}SM22\alpha Cre$, and $Apoe^{-/-}Lmna^{LCS/LCS}$) were sacrificed by $CO_2$ inhalation, and thoracic aortas were extracted, cleaned of fatty tissue, and digested with 2 mg/ml collagenase

(CLS-2, Worthington) for 10 min at 37°C to separate medial and adventitial tissue. Medial aortas from 3 to 5 mice of the same genotype were pooled and snap frozen. Samples were disrupted using TissueLyser (Qiagen), and total RNA was isolated with QIAzol (Qiagen). RNA integrity was confirmed by RNA electrophoresis and with an Agilent 2100 Bioanalyzer.

### Library preparation and RNA sequencing

Total RNA (500 ng) was used to prepare barcoded RNA sequencing libraries using the TruSeq RNA sample preparation kit v2 (Illumina) as described before (Villa-Bellosta *et al*, 2017). In brief, poly-A$^+$ RNA was isolated using poly-T oligo-attached magnetic beads followed by fragmentation and first and second cDNA strand synthesis. cDNA 3′ ends were adenylated and the adapters were ligated. After PCR library amplification, its size was tested using the Agilent 2100 Bioanalyzer DNA 1000 chip and concentration was measured in a Qubit fluorometer (Life Technologies). Libraries were sequenced on a HiSeq2500 sequencer (Illumina), 60-base single reads were generated. FastQ files for each sample were created using CASAVA v1.8 (Illumina). Next-generation sequencing was performed by the CNIC Genomics Unit.

### Differential expression analysis

Sequencing reads were pre-processed by means of a pipeline that used FastQC (Andrews, 2010) to assess read quality, and Cutadapt (Martin, 2011) to trim sequencing reads eliminating Illumina adaptor remains, and to discard reads that were shorter than 30 base pairs. The number of reads obtained per sample was in the range of 8–14 million. The resulting reads were mapped against the mouse transcriptome (GRCm38, release 76; aug2014 archive) and quantified using RSEM v1.17 (Li & Dewey, 2011). The percentage of aligned reads was in the range of 79–82%. Data were then processed with a differential expression analysis pipeline that used the Bioconductor package EdgeR (Robinson *et al*, 2010) for normalization and differential expression testing. Only genes with at least 1 count per million in at least four samples (13,664 genes) were considered for statistical analysis. Three comparisons were made to identify differentially expressed genes in our models: $Apoe^{-/-}$ $Lmna^{G609G/G609G}$ versus $Apoe^{-/-}Lmna^{+/+}$ (Dataset EV1); $Apoe^{-/-}$ $Lmna^{LCS/LCS}SM22\alpha Cre$ versus $Apoe^{-/-}Lmna^{LCS/LCS}$ (Dataset EV2); and $Apoe^{-/-}Lmna^{LCS/LCS}$ versus $Apoe^{-/-}Lmna^{+/+}$ (Dataset EV4). The list of 240 genes shared between Datasets EV1 and EV2 was extracted (Dataset EV3) and logFC (base-2 logarithms of fold change) for those genes were plotted against each other using Microsoft Excel to check for correlation. Differential expression analysis was performed in the CNIC Bioinformatics Unit. Area-proportional Venn diagrams were generated using BioVenn (Hulsen *et al*, 2008) to visualize the overlap between data sets. Overrepresentation test for GO cellular compartment was performed using PANTHER (www.pantherdb.org, release 2017-04-13; GO Ontology database released 2017-08-14).

### Pathway analysis

Ingenuity Pathway Analysis (IPA, Qiagen) was used for more comprehensive transcriptomic data analysis. Briefly, core analyses were performed for the three comparisons in order to visualize pathways altered by progerin expression as well as by the lack of lamin A. Benjamini–Hochberg correction of the *P*-value was applied to extract the significantly altered pathways and stacked bar charts were exported. A comparison analysis was performed to compare results obtained with ubiquitous and VSMC-specific progeroid models. Heatmaps showing canonical pathways and upstream regulators were exported. For more detailed information about IPA tools (Global Canonical Pathways, Upstream regulators, and Comparative analysis), see www.qiagenbioinformatics.com/products/ingenuity-pathway-analysis.

### mRNA isolation and reverse transcription for animal studies

Tissues were homogenized using TissueLyser (Qiagen), and total RNA was extracted with QIAzol reagent (Qiagen). RNA was dissolved in RNase-free water, and concentration was quantified using a NanoDrop spectrophotometer (Wilmington, USA). RNA (1–2 μg) was reverse-transcribed into cDNA using the High Capacity cDNA Reverse Transcription Kit (Applied Biosystems).

### PCR detection of lamin A and progerin

Discrimination between lamin A and progerin mRNAs was performed according to a protocol modified from Yang *et al* (2008). Briefly, cDNA (100 ng) was amplified by PCR using DNA polymerase (Biotools, Spain) and PCR products were separated on a 2% agarose gel with ethidium bromide. Images were taken using a Molecular Imager Gel Doc XR+ System (Bio-Rad) with Image Lab software (Bio-Rad).

### Quantitative real-time PCR for animal studies

Quantitative real-time PCR was performed using Power SYBR Green PCR Master Mix (Applied Biosystems). PCR mixes were loaded on 384-well plates (Applied Biosystems) and run on a 7900-FAST-384 thermal cycler (Applied Biosystems). All the values were normalized to the internal controls beta glucuronidase (*Gusb*) and hypoxanthine guanine phosphoribosyl transferase (*Hprt*) genes. All reactions were performed in triplicate. Primers used in this study are shown in Appendix Table S2.

### Cell culture

Immortalized cell lines were obtained by retroviral introduction of TERT, V12-HRAS, and SV40 large and small T antigens in primary dermal fibroblasts from HGPS patient (NIA Aging Cell Culture Repository, Coriell Institute, AG06297) and age-matched control wild-type individual (CRL-1474 from ATCC) as previously described (Hahn *et al*, 1999; Scaffidi & Misteli, 2011; Fernandez *et al*, 2014). Typical passage number of primary cells before transformation was between 10 and 15, and further experiments in immortalized cells were performed within the first 15 passages after transformation. Cells were grown in MEM containing 15% fetal bovine serum, 2 mmol/l L-glutamine, 100 U/ml penicillin, and 100 μg/ml streptomycin, at 37°C in 5% $CO_2$.

### RNA extraction and reverse transcription for cell studies

RNA was extracted from cells using the RNeasy Mini Kit (Qiagen) according to the manufacturer instructions. RNA (1 μg) was reverse-transcribed into cDNA using iScript cDNA synthesis kit (Bio-Rad) for 20 min at 46°C, after a denaturation step of 1 min at 95°C.

### Quantitative real-time PCR for cell studies

Quantitative real-time PCR was performed using iQ SYBR Green Supermix (Bio-Rad) in a C1000 Touch Thermal Cycler (Bio-Rad). Reaction conditions were as follows: 3 min at 95°C, 1 cycle; 20 s at 95°C, 30 s at 61°C, 40 cycles. All the values were normalized to the internal control glyceraldehyde 3-phosphate dehydrogenase (*GAPDH*) gene. All reactions were performed in triplicate. Primer combinations are indicated in Appendix Table S3.

### Chemical chaperone treatment *in vivo*

TUDCA (580549, Calbiochem) was dissolved in PBS and passed through a 0.22-μm filter. After genotyping, mice were randomized into treatment and control groups (equal or similar number of animals from each litter was assigned to each group). TUDCA (400 mg/kg) or PBS was administered intraperitoneally 3 times a week or for 7 consecutive days, depending on the experiment. Treatment started at 6 weeks of age in the case of $Apoe^{-/-}Lmna^{G609G/G609G}$ mice (and control $Apoe^{-/-}Lmna^{+/+}$ mice) or 8 weeks of age in the case of $Apoe^{-/-}Lmna^{LCS/LCS}SM22\alpha Cre$ mice (and control $Apoe^{-/-}Lmna^{LCS/LCS}$ mice). Treatment ended at 9 weeks of age for gene expression studies, and at 16 weeks of age for high-fat diet experiments. In case of longevity studies, treatment was maintained until animal's death.

For atherosclerosis experiments, animals were fed high-fat diet (10.7% total fat, 0.75% cholesterol, S9167-E010, SSNIFF, Germany) for 8 weeks starting at 8 weeks of age, and sacrificed at 16 weeks of age after overnight fasting.

For longevity experiments, normal chow-fed mice were weighted once a week and inspected for health issues and survival three times a week starting at 6 or 8 weeks of age depending on genotype. One animal was excluded from the study due to health issues associated with injection. Animals that met humane endpoint criteria were sacrificed and the deaths recorded.

### Quantification of atherosclerosis burden

Aortic atherosclerosis was quantified as previously described (Hamczyk *et al*, 2018b). Briefly, mouse aortas were fixed with 4% formaldehyde in PBS, cleaned of fatty tissue, and stained with 0.2% Oil Red O (ORO, O0625, Sigma). The thoracic aorta was cut open longitudinally and pinned out flat for planimetric analysis. Images were taken with a digital camera (OLYMPUS UC30) mounted on a stereo microscope (OLYMPUS SZX3). Lesion area was quantified as a percentage of ORO-positive aortic surface using SigmaScan Pro 5 software (Systat Software Inc) by a researcher blinded to genotype.

### Immunofluorescence and histology

Organs and embryos were fixed with 4% formaldehyde in PBS and either included in Tissue-Tek OCT compound (SAKURA, Netherlands) or paraffin. For OCT-embedded specimens (aortic arch), 8-μM sections were prepared and stained with hematoxylin-eosin (H&E). For immunofluorescence, sections were incubated for 1 h at room temperature (RT) in blocking and permeabilizing solution: PBS containing 0.3% Triton X-100 (9002-93-1, Sigma), 5% normal goat serum (005-000-001, Jackson ImmunoResearch), and 5% bovine serum albumin (BSA, A7906, Sigma). Sections were then incubated overnight at 4°C with anti-BiP (3177, Cell Signaling, 1:100), anti-PDI (3501, Cell Signaling, 1:100), and anti-calreticulin (ab92516, abcam, 1:250) antibodies diluted in PBS containing 0.3% Triton X-100 and 2.5% normal goat serum. Samples were incubated for 2 h at RT with an anti-α-smooth muscle actin-Cy3 (SMA-Cy3, C6198, Sigma, 1:200) antibody, a secondary antibody (A-21245, goat anti-rabbit Alexa Fluor 647, Invitrogen), and nucleic acid stain Hoechst 33342 (B2261, Sigma), and mounted in Fluoromount G imaging medium (00-4958-02, Affymetrix eBioscience).

For paraffin-embedded specimens (embryo, aortic arch, thoracic aorta, aortic root, kidney, duodenum, jejunum, ileum, and colon), 4-μM sections were prepared. For immunofluorescence, antigen retrieval was performed with 10 mM sodium citrate buffer (pH 6). Samples were blocked for 1 h at RT with 5% BSA and 5% normal goat serum in PBS. Sections were incubated overnight at 4°C with anti-BiP (3177, Cell Signaling, 1:100), anti-PDI (3501, Cell Signaling, 1:100), anti-calreticulin (ab92516, abcam, 1:250), anti-lamin A/progerin (sc-20680, Santa Cruz Biotechnology, 1:100), and anti-CD31 (DIA-310, Dianova, 1:50) antibodies diluted in PBS containing 2.5% normal goat serum. Sections were incubated for 2 h at RT with an anti-SMA-Cy3 antibody (C6198, Sigma, 1:200), secondary antibodies (A-21245, goat anti-rabbit Alexa Fluor 647, Invitrogen, and A-11006, goat anti-rat Alexa Fluor 488, Invitrogen), and/or Hoechst 33342 stain (B2261, Sigma) diluted in 2.5% normal goat serum in PBS, and mounted using Fluoromount G imaging medium.

H&E-stained sections were scanned using a NanoZoomer-RS scanner (Hamamatsu), and images were exported with NDP.view2. Fluorescence images were acquired with a Zeiss LSM 700 confocal microscope. Image analysis was performed using ImageJ Fiji software by a researcher blinded to genotype. Smooth muscle content (as % of SMA-positive area) in the muscularis externa of the intestine was quantified in one section of duodenum, jejunum, ileum, and colon per animal. Aorta and plaque features were analyzed in approximately 3 sections (for aortic arch and aortic root) and/or 4 (for thoracic aorta) per animal, and the mean was used for the statistical analysis.

### Statistical analysis

Kolmogorov–Smirnov and D'Agostino-Pearson normality tests were used to evaluate data distribution. For parametric data, a *t*-test was used to compare two groups (two-sample *t*-test) or one group to a hypothetical value (one-sample *t*-test). Statistical tests were one- or two-tailed depending on the previous hypothesis (e.g., one-tailed *t*-test was used for RNAseq data validation experiments). For two groups with unequal variances, a Welch's correction was applied. To compare multiple groups, a one-way ANOVA with Tukey's *post hoc* test was used, as this approach is resistant to normality violations and unequal variances with similar-sized groups. For Kaplan–Meier survival curves, a log-rank (Mantel-Cox) test was applied. Data were represented as mean (±SEM). Outliers were evaluated

## The paper explained

### Problem
Hutchinson–Gilford progeria syndrome (HGPS) is a premature aging syndrome caused by expression of an aberrant protein called progerin. Patients present accelerated atherosclerosis and die from myocardial infarction or stoke in their teens, despite having normal serum cholesterol levels. One of the most striking characteristics of HGPS is the massive loss of vascular smooth muscle cells (VSMC) in the vessel wall. However, the mechanisms leading to progerin-driven VSMC death remain largely unexplored. Furthermore, there is no effective treatment targeting progerin-induced atherosclerosis, the death-causing symptom in HGPS.

### Results
High-throughput transcriptomic analysis of pre-disease medial aortas from ubiquitous and VSMC-specific progeroid mice identified endoplasmic reticulum (ER) stress and the subsequent unfolded protein response (UPR) as possible drivers of VSMC death. Activation of this pathway was also detected in HGPS patient-derived cells. Treatment with tauroursodeoxycholic acid (TUDCA), a chemical chaperone that increases the ability of a cell to tolerate ER stress, delayed VSMC loss and inhibited atherosclerotic plaque formation in the aorta of both progeroid models fed high-fat diet. Moreover, TUDCA prolonged survival in the normal chow-fed VSMC-specific progeroid model.

### Impact
This study identifies ER stress and the UPR as a novel pathway in the etiopathology of HGPS, especially in relation to VSMCs. The study furthermore proposes the use of TUDCA to treat progerin-induced vascular disease. Because progerin has been found at low levels in cells and tissues from normally aging individuals, the results of the present study may also be relevant to research on physiological aging and associated cardiovascular disease. Moreover, our findings may shed light on other diseases involving VSMC death.

using GraphPad outlier calculator (www.graphpad.com/quickcalcs/Grubbs1.cfm). GraphPad Prism 5 was used for statistical analysis. Differences were considered significant at $P < 0.05$.

## Data Availability

RNA sequencing data were deposited in the NCBI SRA, accession number: SRP099105 (https://www.ncbi.nlm.nih.gov/sra/?term = SRP099105). Other data that support the findings of this study are available from the corresponding authors upon request.

**Expanded View** for this article is available online.

## Acknowledgements
We thank Simon Bartlett for English editing, the CNIC Genomics and Bioinformatics Units for RNAseq studies and support with RNAseq data analysis, and Eva Santos, Virginia Zorita, and the CNIC Animal Facility for animal care. Work in VA's laboratory is supported by grants from the Spanish Instituto de Salud Carlos III (RD12/0042/0028, AC17/00067 and AC16/00091) and Ministerio de Ciencia, Innovación y Universidades (MCIU) (SAF2016-79490-R), with co-funding from the Fondo Europeo de Desarrollo Regional (FEDER, "Una manera de hacer Europa"), the Progeria Research Foundation (Established Investigator Award 2014-52), and the Fundació Marató TV3 (122/C/2015). The CNIC is supported by the MCIU and the Pro CNIC Foundation, and is a Severo Ochoa Center of Excellence (SEV-2015-0505). Work in the Lopez-Otin laboratory is supported by grants from the Ministerio de Economía y Competitividad (MINECO/FEDER), the European Research Council, and the Progeria Research Foundation. The Instituto Universitario de Oncología is supported by Obra Social Cajastur. Work in the Misteli laboratory was supported by the Intramural Research Program of the National Institutes of Health, National Cancer Institute, and Center for Cancer Research. The MCIU supported RVB ("Juan de la Cierva" JCI-2011-09663 and SAF-2014-60699-JIN postdoctoral contracts) and MRH (FPI predoctoral contract BES-2011-043938 and "Juan de la Cierva" FJCI-2017-33299 postdoctoral contract). RMN is supported by the Ministerio de Educación, Cultura y Deporte (FPU predoctoral contract FPU16/05027). VQ is supported by grants from the Principado de Asturias and MINECO, including FEDER funding. SV was supported by an Erwin Schroedinger Fellowship from the Austrian Science Fund (J3849-B28).

## Author contributions
MRH designed and performed experiments, analyzed data, and prepared figures and the manuscript. RV-B supervised research and project planning, designed and performed experiments, and analyzed data. VQ, PG, SV, RMN, and MJA-M performed experiments and analyzed data. TM provided advice on the design, supervision, and analysis of experiments. CL-O supervised project planning and data interpretation. VA supervised the research, project planning, and data interpretation and prepared the manuscript. All authors read and approved the manuscript.

## Conflict of interest
The authors declare that they have no conflict of interest.

## For more information
(i)   Progeria Research Foundation: www.progeriaresearch.org
(ii)  Hutchinson–Gilford progeria syndrome: www.omim.org/entry/176670
(iii) Lamin A/C: www.omim.org/entry/150330

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
