## [Review Process File · EMBO Molecular Medicine]

Progerin accelerates atherosclerosis by inducing endoplasmic reticulum stress in vascular smooth muscle cells

Magda R. Hamczyk, Ricardo Villa-Bellosta, Víctor Quesada, Pilar Gonzalo, Sandra Vidak, Rosa M. Nevado, María J. Andrés-Manzano, Tom Misteli, Carlos López-Otín, and Vicente Andrés

Review timeline:

Submission date:	29 August 2018
Editorial Decision:	21 September 2018
Revision received:	7 January 2019
Editorial Decision:	18 January 2019
Revision received:	11 February 2019
Accepted:	13 February 2019

Editor: Lise Roth

Transaction Report:

1st Editorial Decision

21 September 2018

Thank you for the submission of your manuscript to EMBO Molecular Medicine. We have now heard back from the three referees whom we asked to evaluate your manuscript.

As you will see from the reports below, the three referees are positive and support publication of the article in EMBO Molecular Medicine pending appropriate revisions. Addressing the reviewers concerns in full will be necessary for further considering the manuscript in our journal. EMBO Molecular Medicine encourages a single round of revision only and therefore, acceptance or rejection of the manuscript will depend on the completeness of your responses included in the next, final version of the manuscript.

EMBO Molecular Medicine has a "scooping protection" policy, whereby similar findings that are published by others during review or revision are not a criterion for rejection. Should you decide to submit a revised version, I do ask that you get in touch after three months if you have not completed it, to update us on the status. Please also contact us as soon as possible if similar work is published elsewhere. If other work is published, we may not be able to extend the revision period beyond three months.

I look forward to receiving your revised manuscript.

***** Reviewer's comments *****

Referee #1 (Remarks for Author):

The manuscript by Hamczyk et al., entitled "Progerin accelerates atherosclerosis by inducing endoplasmic reticulum stress in vascular smooth muscle cells", investigates one of the main

pathophysiological mechanism, the loss of VSMC, leading to heart attack and stroke in progeria patients that lead to their premature death.

The study compares two C57BL/6J male mice models with ubiquitous and VSMC-specific (Apoe^{-/-} LmnaLCS/LCSSM22 α Cre) progerin expression and their littermate controls, already described in previous publications by the authors.

Apolipoprotein E-deficient (Apoe^{-/-}) mice is a widely used atherosclerosis model after high fat diet (HFD) triggering, as done in the present study.

The main results can be summarized as following :

1. A RNAseq analysis confirmed by PCR compared gene expression between aortic media dissected out from mice expressing progerin either ubiquitously (Apoe^{-/-} LmnaG609G/G609G) or restricted in cells expressing transgelin encoded by SM22a gene (Apoe^{-/-} LmnaLCS/LCSSM22 α Cre). Five pathways, hepatic fibrosis/hepatic stellate cell activation, NRF2-mediated oxidative stress response, UPR, ER stress pathway and RAN signaling exhibited significant differences. The manuscript was then focussed on two pathways only, UPR and ER stress.

The RAN pathway, that evidenced the widest difference, was not addressed in this study, even if it is well known that progeria cells exhibit nucleo-cytoplasmic exchange abnormalities.

2. 6 genes encoding either ER resident proteins or a transcription factor derived from an ER protein showed an increased expression in human progeria cell cultures, in medial aorta from mice models and at least in kidney and heart from Apoe^{-/-} LmnaG609G/G609G mice ubiquitously expressing progerin.

3. The intraperitoneal administration of TUDCA (tauroursodeoxycholic acid, a low abundance bile acid in humans) reduced the VSMC loss et atherosclerotic lesions in medial aorta from HFD-fed mice models expressing progerin either in all cells or in SM22 α expressing cells only.

4. TUDCA administration increased life span of Apoe^{-/-} LmnaLCS/LCSSM22 α Cre mice expressing progerin in SM22 α expressing cells, but not in Apoe^{-/-} LmnaG609G/G609G mice whose progerin expression is ubiquitous.

The manuscript gives strong and convincing RNAseq and PCR data demonstrating abnormalities in UPR and ER stress signaling pathways in cultured human progeria cells, in cells from mice medial aorta expressing progerin and transgelin encoded by SM22a gene.

Convincing data are also provided regarding the efficacy of TUDCA in lowering medial aorta VSMC loss et atherosclerotic lesions in Apoe^{-/-} and HFD-fed mice expressing progerin, either in all cells or in SM22 α expressing cells, the efficacy of TUDCA in improving the life span only of mice expressing progerin in SM22 α positive cells.

However several data are missing in the manuscript :

1. The study does not provide data showing the correction of UPR/ER stress signaling pathways by TUDCA, that could support the use of TUDCA in the treatment of vascular disorders encountered in progeria, in light of the data showing no lifespan improvement in Apoe^{-/-} LmnaG609G/G609G mice where progerin expression is ubiquitous.

2. SM22 α encoding transgelin is expressed in other cells than VSMC, as for example in smooth muscle cells within digestive tract, in cells from kidney glomerulus... Depending upon the size of SM22a promoter used, the expression of transgenes under the control of SM22a promoter has been shown to be ubiquitous in smooth muscle cells in mice embryos, and restricted to VSMC in adult mice.

Did the authors search for progerin expression in digestive tract from Apoe^{-/-} LmnaLCS/LCSSM22 α Cre mice ?

Did these transgelin expressing adult cells outside vascular wall disappear when expressing progerin ?

Does SM22a-controlled progerin expression in mice embryonic development could lead to smooth muscle cells loss in vascular wall and/or in other visceral organs ?

3. The manuscript does not explore the pathophysiological mechanism(s) resulting in VSMC loss. However, the mice models designed by the authors could be useful in exploring the hypotheses regarding VSMC loss in progeria and reported in a recent review by some of the authors of the present study.

4. Besides triggering atherosclerosis in mice models, HFD has been shown to induce both changes in gut microbiota and in intestinal permeability, inflammatory events, metabolic parameters... in mice, that could be worsened if progerin leads to gut smooth muscle cells loss.

In conclusion, the present manuscript provides very important data, but requires at least two supplemental informations, before to be published in EMBO Molecular Medicine :

1. The evidence that TUDCA corrected UPR/ER stress pathway in mice model
2. The control that progerin does not result in smooth muscle cell loss in other organs than vascular wall, either during embryonic development or in adult.

The discussion section would be improved by addressing the above points.

Referee #2 (Remarks for Author):

The manuscript by Magda R. Hamczk et al. entitled "Progerin accelerates atherosclerosis by inducing endoplasmic reticulum stress in vascular smooth muscle cells" provides a new molecular mechanism of the development of the vascular disease in HGPS.

This is the first study investigating the molecular alterations caused by progerin accumulation in VSMCs and its effect on atherosclerosis in vivo. This work is significant and the findings might be relevant to other syndromes causing vascular disease.

Comments on the manuscript:

In a previous study, the team demonstrated that progerin-driven vascular smooth muscle cell (VSMC) loss accelerates atherosclerosis leading to premature death in apolipoprotein E-deficient mice. Now in this present work they took the task to identify the molecular mechanism underlying this process.

To address this question they conducted a transcriptomic analysis of progerin-expressing aortas using two mouse models of HGPS.

One express progerin ubiquitously (Apoe^{-/-}LmnaG609G/G609G), the other progerin expression is restricted to VSMCs (Apoe^{-/-}LmnaLCS/LCSSM22aCre).

Next. They compare two time points: prior signs of atherosclerosis (aortas from 8 weeks old mice) and at the onset of atherosclerosis (aortas from 16 weeks old mice).

1) The results presented in Figure 1 on the differential expression analysis of RNA sequencing (Fig. 1E) show four common signaling between VSMC-specific and ubiquitous groups. However, the RAN signaling seems highly altered in VSMC-specific while not affected in ubiquitous. Because RAN signaling has been implicated in other HGPS studies, it will be important to validate these results in ubiquitous group. For instance quantify by real-time PCR a few genes from this pathway.

2) The RNA sequencing results are in part validated by quantitative real-time PCR. A set of ER stress response and UPR genes show a significant upregulation in aortic media from 8-week-old mice in both progeria models (Fig. 2). Importantly, these genes are not altered in tissues that do not express progerin in Apoe^{-/-}LmnaLCS/LCSSM22aCre mice. This result provides convincing evidence supporting the implication of progerin in activation of these two pathways. However, whether it is a direct or indirect effect of progerin remains to be demonstrated at the molecular level. The authors screened the same set of genes in immortalized human HGPS cells (Fig. 2 D). They state on Page 4: "We next used quantitative real-time PCR to analyze immortalized HGPS patient-derived cells."

I simply want to check whether indeed they used immortalized or primary fibroblasts lines. If immortalized cells were used please comment of this choice?

Next, it has now been established that the level of progerin expression in HGPS cells is critical for inducing cellular malfunctions and most likely for ER stress. So, it would be important to show a progerin or lamin A/C western blot detection of HGPS cell extracts used in this study. Hence, progerin expression levels is variable in different tissues in HGPS mice and might as well influence the difference observed in gene expression profiles.

3) The immunocytochemistry staining (Fig. EV3) support the conclusion that there is an increase ER stress as indicated in aortas sections from mice models expressing progerin. It is a pity that there is no mouse specific anti-progerin antibody available to visualize this effector over these markers.

4) To further assess the implication of ER stress response and UPR signaling in VSMC death and atherosclerosis in the ubiquitous and VSMC- specific progeria mice. The authors tested TUDCA, a chemical chaperone.

The Chromatin and SMA staining of aorta sections convincingly support their conclusion that TUDCA treatment alleviates aortic VSMC loss in both mice models (Fig.3). Consequently, TUDCA treatment seems to prevent atherosclerosis development.

There is one sticking observation for the significant amelioration of the vascular system; the lifespan of Apoe^{-/-}LmnaG609G/G609G mice is not increased. The authors refer to their previous publication (Hamczyk et al, 2018b) indicating that they die from different causes than vascular disease. A summary of the reasons causing their early death, independent of vascular disease will be helpful to be included in this publication. This information might be critical in vascular disease of individuals suffering from other syndromes than HGPS.

Finally, this study reports very fascinating and important findings on the potential link between progerin-ER stress and vascular disease, the most critical pathology of HGPS children. I am sure this study will attract readers from various fields beside HGPS.

Referee #3 (Remarks for Author):

The same group has recently reported the generation of mouse models exhibiting progerin-induced acceleration of atherosclerosis, similarly to HFPS patients, identifying VSMCs as a major target for progerin in the vessel wall (Circulation 2018). Here, the authors performed RNA seq of the tunica media from APOE-KO mouse aorta with ubiquitous and VSMC-specific progerin expression with the aim of identifying progeria-induced VSMC-death and accelerated atherosclerosis. Results of high-throughput sequencing indicated that ER stress response and associated UPR gene expression are significantly affected in progerin-expressing VSMCs. Similar, these pathways were found altered in fibroblasts of HGPS patients. The authors used TUDCA for alleviating ERstress/UPR in their models, demonstrating a delayed VSMC loss and reduced atherosclerosis, importantly with life-span prolongation.

The study is well conducted. I have only some minor considerations:

- a. Quantification of immunostaining in Figure EV3 should be provided. I suggest also to move the figure to the main manuscript as it includes relevant data.
- b. The same authors in a recent Circulatio manuscript demonstrated that atherosclerosis increase in mice expression progerin is not associated with increased serum cholesterol levels. Could this point be commented? Does it have implication for data interpretation of this manuscript?
- c. For helping reading to an audience not familiar with progeria mouse models, I suggest to include in the M&M section a brief description of the Lmna LCSL/LCS mouse model used as control in some experiments.

1st Revision - authors' response

7 January 2019

(See next page)

RESPONSE TO EDITORIAL DECISION LETTER AND TO REFEREE'S COMMENTS
RE: EMM-2018-09736

To prepare the revised manuscript, we have followed all the instructions in the editorial decision letter of September 21, 2018. We are grateful to the Reviewers for their enthusiasm for our study and for their helpful comments, which have helped to significantly improve our work. Below, we provide a point-by-point response to the Reviewers' comments. At the end of this document, we include 10 additional figures for the Reviewers (RV1-10) and the cited references. To facilitate the review process, we have submitted a revised version of the manuscript with new text in red font.

Referee #1 (Remarks for Author):

The manuscript by Hamczyk et al., entitled "Progerin accelerates atherosclerosis by inducing endoplasmic reticulum stress in vascular smooth muscle cells", investigates one of the main pathophysiological mechanisms, the loss of VSMC, leading to heart attack and stroke in progeria patients that lead to their premature death. The study compares two C57BL/6J male mice models with ubiquitous and VSMC-specific (*Apoe*^{-/-}*Lmna*^{LCS/LCSSM22 α Cre) progerin expression and their littermate controls, already described in previous publications by the authors. Apolipoprotein E-deficient (*Apoe*^{-/-}) mice is a widely used atherosclerosis model after high fat diet (HFD) triggering, as done in the present study. The main results can be summarized as following:}

1. A RNAseq analysis confirmed by PCR compared gene expression between aortic media dissected out from mice expressing progerin either ubiquitously (*Apoe*^{-/-} *Lmna*^{G609G/G609G}) or restricted in cells expressing transgelin encoded by *SM22 α* gene (*Apoe*^{-/-}*Lmna*^{LCS/LCSSM22 α Cre). Five pathways, hepatic fibrosis/hepatic stellate cell activation, NRF2-mediated oxidative stress response, UPR, ER stress pathway and RAN signaling exhibited significant differences. The manuscript was then focussed on two pathways only, UPR and ER stress. The RAN pathway, that evidenced the widest difference, was not addressed in this study, even if it is well known that progeria cells exhibit nucleo-cytoplasmic exchange abnormalities.}

2. 6 genes encoding either ER resident proteins or a transcription factor derived from an ER protein showed an increased expression in human progeria cell cultures, in medial aorta from mice models and at least in kidney and heart from *Apoe*^{-/-} *Lmna*^{G609G/G609G} mice ubiquitously expressing progerin.

3. The intraperitoneal administration of TUDCA (taurooursodeoxycholic acid, a low abundance bile acid in humans) reduced the VSMC loss et atherosclerotic lesions in medial aorta from HFD-fed mice models expressing progerin either in all cells or in *SM22 α* expressing cells only.

4. TUDCA administration increased life span of *Apoe*^{-/-}*Lmna*^{LCS/LCSSM22 α Cre mice expressing progerin in *SM22 α* expressing cells, but not in *Apoe*^{-/-} *Lmna*^{G609G/G609G} mice whose progerin expression is ubiquitous.}

The manuscript gives strong and convincing RNAseq and PCR data demonstrating abnormalities in UPR and ER stress signaling pathways in cultured human progeria cells, in cells from mice medial aorta expressing progerin and transgelin encoded by *SM22 α* gene.

Convincing data are also provided regarding the efficacy of TUDCA in lowering medial aorta VSMC loss et atherosclerotic lesions in *Apoe*^{-/-} and HFD-fed mice expressing progerin, either in all cells or in *SM22 α* expressing cells, the efficacy of TUDCA in improving the life span only of mice expressing progerin in *SM22 α* positive cells.

However several data are missing in the manuscript:

1. The study does not provide data showing the correction of UPR/ER stress signaling pathways by TUDCA, that could support the use of TUDCA in the treatment of vascular disorders encountered in progeria, in light of the data showing no lifespan improvement in *Apoe*^{-/-}*Lmna*^{G609G/G609G} mice where progerin expression is ubiquitous.

RESPONSE: We have included a new experiment, in which 8-week-old mice were treated with TUDCA for 7 consecutive days, sacrificed and the ER stress/UPR pathway was analyzed by qPCR in medial aortas. We have used *Apoe*^{-/-}*Lmna*^{LCS/LCS}*SM22αCre* mice to avoid the systemic effects of progerin expression in *Apoe*^{-/-}*Lmna*^{G609G/G609G} mice. Various studies have shown that TUDCA augments the capacity of a cell to sustain ER stress rather than to block it (Uppala et al, 2017; Vandewynckel et al, 2015), in other words, TUDCA can mitigate the pro-apoptotic UPR. In line with these observations, we found that TUDCA slightly increased the expression of some ER stress-related genes and decreased the expression of *Ddit3* gene, coding for a key transcription factor that switches between survival and death during ER stress. These results, which indicate that TUDCA prolongs the lifespan of progerin-expressing VSMCs by maintaining the cell in a pro-survival phase of the ER stress-UPR response, have been included in the new **Figure EV3**, and are described in the Results section (page 5, second paragraph); this information has been also included in the Synopsis.

Regarding the survival of *Apoe*^{-/-}*Lmna*^{G609G/G609G} mice with ubiquitous progerin expression, we have previously demonstrated that these mice typically do not die from atherosclerosis complications (Hamczyk et al, 2018); thus, a treatment targeting this pathology is not expected to extend lifespan of these animals. The possible causes of death in *Apoe*^{-/-}*Lmna*^{G609G/G609G} mice have been added in the Discussion section (page 7, first paragraph).

2. SM22α encoding transgelin is expressed in other cells than VSMC, as for example in smooth muscle cells within digestive tract, in cells from kidney glomerulus... Depending upon the size of SM22a promoter used, the expression of transgenes under the control of SM22a promoter has been shown to be ubiquitous in smooth muscle cells in mice embryos, and restricted to VSMC in adult mice.

Did the authors search for progerin expression in digestive tract from *Apoe*^{-/-}*Lmna*^{LCS/LCS}*SM22αCre* mice ?

Did these transgelin expressing adult cells outside vascular wall disappear when expressing progerin?

Does SM22α-controlled progerin expression in mice embryonic development could lead to smooth muscle cells loss in vascular wall and/or in other visceral organs?

RESPONSE: As suggested by the Reviewer, we have examined progerin expression in *Apoe*^{-/-}*Lmna*^{LCS/LCS}*SM22αCre* embryos at 11.5 and 15.5 days post coitum (dpc) (**Figures RV1-4**). *Apoe*^{-/-}*Lmna*^{LCS/LCS} embryos (from the same litter) expressing only lamin C were used as a negative control. At 11.5 dpc, progerin expression in *Apoe*^{-/-}*Lmna*^{LCS/LCS}*SM22αCre* embryos was undetectable in the aorta (**Figure RV1A**), intestine (**Figure RV3A**) and kidney (**Figures RV4A**), and very low level of expression was observed in some cells of the heart (**Figure RV2A**). At 15.5 dpc, we found progerin expression in SMCs of the aortic media (**Figure RV1B**) and to some extent in the aortic adventitia (fibroblasts, **Figure RV1B**), in agreement with our previous results in adult mice (Hamczyk et al, 2018). As expected, we also found expression of progerin in the heart, predominantly evident in the valves, with some expression also present in the ventricle wall (**Figure RV2B**). At 15.5 dpc, progerin expression in the intestine was limited to a few cells in the muscularis, with no progerin-positive cells in the mucosa (**Figure RV3B**). Expression of progerin in the kidney of *Apoe*^{-/-}*Lmna*^{LCS/LCS}*SM22αCre* embryos was undetectable or extremely low (**Figures RV4B**). Moreover, we did not observe any evident loss of SMCs in any of the organs tested. It is important to note that SMC loss in the aorta of *Apoe*^{-/-}*Lmna*^{LCS/LCS}*SM22αCre* mice was not evident until approximately 8 weeks of age, thus it is unlikely that progerin expression in the embryo could cause their loss.

These results are consistent with previous studies showing that lamin A expression is extremely low during development (it is first detected in the 12 dpc embryo) and increases upon cell differentiation (Prather et al, 1991; Rober et al, 1989; Schatten et al, 1985; Stewart & Burke, 1987). Moreover, various studies suggest a limited role of lamin A during embryonic development, e.g. *Lmna* knockout mice exhibit normal embryonic development, but start to show growth retardation and muscular problems shortly after birth, and die by 8 weeks of age (Sullivan et al, 1999). Accordingly, *Lmna*^{G609G/G609G} mice with ubiquitous progerin expression look normal at birth and appear healthy until ~3 weeks of age; however, later on begin to display premature ageing features and die at 15 weeks of age on average. Likewise, HGPS patients are normal at birth and are typically not diagnosed until 1.6-2.9 years of age (Hennekam, 2006; Merideth et al, 2008), confirming limited effects of progerin during embryonic development.

We have explored progerin expression in the kidney and intestine (duodenum, jejunum, ileum and colon) of 12- and 26-week-old *Apoe*^{-/-}*Lmna*^{LCS/LCS}*SM22aCre* mice (**Figures RV5-7**). *Apoe*^{-/-}*Lmna*^{LCS/LCS} littermates expressing only lamin C were used as a negative control. Kidneys of *Apoe*^{-/-}*Lmna*^{LCS/LCS}*SM22aCre* mice showed progerin expression predominantly restricted to arteries and arterioles, with low expression of the mutant protein in sparse cells outside the vessel wall (**Figure RV5**). Progerin expression in the intestine of *Apoe*^{-/-}*Lmna*^{LCS/LCS}*SM22aCre* mice was detected principally in the muscularis externa (**Figure RV6A, C**) and was not accompanied by SMC loss in any of the regions analyzed in both the circular and the longitudinal muscle (**Figure RV6B, D**). We have also observed some progerin expression in the mucosa, especially evident in the ileum (**Figure RV6A, C**). Progerin expression in the intestine was mosaic-like, so only some cells expressed the mutant protein. In agreement with previous findings showing that mosaic mice expressing prelamin A (an unprocessed form of lamin A exhibiting toxic properties similar to progerin) have no progeroid phenotype (de la Rosa et al, 2013), we did not observe any pathological changes in the intestine of *Apoe*^{-/-}*Lmna*^{LCS/LCS}*SM22aCre* mice at 12 weeks of age (**Figure RV7**). At 26 weeks of age, far beyond the time points used in the present study, we have only found an increased myenteric plexus area in the colon of *Apoe*^{-/-}*Lmna*^{LCS/LCS}*SM22aCre* mice, which is not pathologic itself. In conclusion, as expected, we have observed some progerin expression in the intestine of *Apoe*^{-/-}*Lmna*^{LCS/LCS}*SM22aCre* mice; however, it did not have any obvious detrimental effects.

3. The manuscript does not explore the pathophysiological mechanism(s) resulting in VSMC loss. However, the mice models designed by the authors could be useful in exploring the hypotheses regarding VSMC loss in progeria and reported in a recent review by some of the authors of the present study.

RESPONSE: The Reviewer has raised a very important point, which will require future studies beyond the scope of the present work. To acknowledge this important issue, we added the following sentence in Discussion (page 6, second paragraph): “Future studies are warranted to assess whether progerin induces ER stress directly or indirectly, and to identify the exact mechanism of ER stress/UPR-related VSMC death triggered by progerin.”

4. Besides triggering atherosclerosis in mice models, HFD has been shown to induce both changes in gut microbiota and in intestinal permeability, inflammatory events, metabolic parameters... in mice, that could be worsened if progerin leads to gut smooth muscle cells loss.

RESPONSE: As shown in **Figure RV6**, we have not observed any SMC loss in the intestine of *Apoe*^{-/-}*Lmna*^{LCS/LCS}*SM22aCre* mice compared to *Apoe*^{-/-}*Lmna*^{LCS/LCS} control mice. Accordingly, we have not observed any significant differences in the leukocyte subpopulations and cholesterol levels between *Apoe*^{-/-}*Lmna*^{LCS/LCS}*SM22aCre* and *Apoe*^{-/-}*Lmna*^{LCS/LCS} mice fed HFD (Hamczyk et al, 2018). It is important to note that except for **Figure 4** in the main manuscript and **Table EV2**, all the experiments in the present study were performed with mice fed normal chow.

In conclusion, the present manuscript provides very important data, but requires at least two supplemental informations, before to be published in EMBO Molecular Medicine:

1. The evidence that TUDCA corrected UPR/ER stress pathway in mice model

RESPONSE: We have included a new **Figure EV3** and a short description in the Results section (please, see **Response to Comment 1**).

2. The control that progerin does not result in smooth muscle cell loss in other organs than vascular wall, either during embryonic development or in adult.

RESPONSE: We have shown that progerin expression is almost undetectable in the embryo at 11.5 dpc and present in the aorta, and to a lesser extent, in the heart and muscularis of the intestine in the embryo at 15.5 dpc. Moreover, progerin expression in the intestine of adult *Apoe^{-/-}Lmna^{LCS/LCS}SM22 α Cre* mice does not cause smooth muscle cell loss in the muscularis externa (please, see **Figures RV1-7** and **Response to Comment 2**).

The discussion section would be improved by addressing the above points.

RESPONSE: As explained in previous responses, we have addressed in the Discussion section the points raised by Reviewer #1.

Referee #2 (Remarks for Author):

The manuscript by Magda R. Hamczk et al. entitled "Progerin accelerates atherosclerosis by inducing endoplasmic reticulum stress in vascular smooth muscle cells" provides a new molecular mechanism of the development of the vascular disease in HGPS.

This is the first study investigating the molecular alterations caused by progerin accumulation in VSMCs and its effect on atherosclerosis in vivo. This work is significant and the findings might be relevant to other syndromes causing vascular disease.

Comments on the manuscript:

In a previous study, the team demonstrated that progerin-driven vascular smooth muscle cell (VSMC) loss accelerates atherosclerosis leading to premature death in apolipoprotein E-deficient mice. Now in this present work they took the task to identify the molecular mechanism underlying this process.

To address this question they conducted a transcriptomic analysis of progerin-expressing aortas using two mouse models of HGPS. One express progerin ubiquitously (*Apoe^{-/-}Lmna^{G609G/G609G}*), the other progerin expression is restricted to VSMCs (*Apoe^{-/-}Lmna^{LCS/LCSSM22 α Cre}*). Next. They compare two time points: prior signs of atherosclerosis (aortas from 8 weeks old mice) and at the onset of atherosclerosis (aortas from 16 weeks old mice).

1) The results presented in Figure 1 on the differential expression analysis of RNA sequencing (Fig. 1E) show four common signaling between VSMC-specific and ubiquitous groups. However, the RAN signaling seems highly altered in VSMC-specific while not affected in ubiquitous. Because RAN signaling has been implicated in other HGPS studies, it will be important to validate these results in ubiquitous group. For instance quantify by real-time PCR a few genes from this pathway.

RESPONSE: As suggested by the Reviewer, we have examined by qPCR the expression of several genes from the RAN pathway in both the ubiquitous and the VSMC-specific progeroid models (**Figure RV8**). As anticipated, medial aortas of *Apoe^{-/-}Lmna^{LCS/LCS}SM22αCre* mice presented increased mRNA levels of *Cse1l*, *Kpna1*, *Kpna3*, *Kpna4*, *Rangap1* and *Rcc1* genes as compared to *Apoe^{-/-}Lmna^{LCS/LCS}* control littermates (**Figure RV8B**). *Xpo1* and *Ran* genes showed strong tendency towards upregulation, but differences did not reach statistical significance (**Figure RV8B**). Compared to *Apoe^{-/-}Lmna^{+/+}* controls, medial aortas of *Apoe^{-/-}Lmna^{G609G/G609G}* mice only showed upregulation of *Kpna1*, *Kpna3* and *Xpo1* (**Figure RV8A**). These results are in agreement with RNAseq data showing considerable alterations in RAN signaling in *Apoe^{-/-}Lmna^{LCS/LCS}SM22αCre* mice, but not in *Apoe^{-/-}Lmna^{G609G/G609G}* mice (**Figure RV8C**). In our experience, the vascular phenotype of *Apoe^{-/-}Lmna^{G609G/G609G}* mice is slightly delayed compared to *Apoe^{-/-}Lmna^{LCS/LCS}SM22αCre* mice, in part due to hindered food intake by *Apoe^{-/-}Lmna^{G609G/G609G}* mice. Since increases in gene expression within RAN pathway were modest in the VSMC-specific model (fold change of approx. 1.2-1.3 for most of the genes; see **Figure RV8C**), it is possible that the ubiquitous model will show more pronounced alterations in RAN signaling at older ages. Indeed, multiple ‘omics’ experiments performed in our laboratory show that with increasing age, mice ubiquitously expressing progerin display alterations in the majority of the expressed genes, and therefore most of the cellular pathways become affected. Thus, in this study we have searched for a primary mechanism that could lead to VSMC death and was evident at a very early stage of the disease in both mouse models.

2) The RNA sequencing results are in part validated by quantitative real-time PCR. A set of ER stress response and UPR genes show a significant upregulation in aortic media from 8-week-old mice in both progeria models (Fig. 2). Importantly, these genes are not altered in tissues that do not express progerin in *Apoe^{-/-}Lmna^{LCS/LCS}SM22αCre* mice. This result provides convincing evidence supporting the implication of progerin in activation of these two pathways. However, whether it is a direct or indirect effect of progerin remains to be demonstrated at the molecular level.

RESPONSE: Since lamin A plays multiple roles in the nucleus, from structural support to transcription regulation and signal transduction, it is an extremely complex task to examine whether progerin induces ER stress in a direct or an indirect manner. This issue is beyond the scope of this study; however, to highlight the importance of addressing this question, we have included the following sentence in the Discussion section (page 6, second paragraph): “Future studies are warranted to assess whether progerin induces ER stress directly or indirectly, and to identify the exact mechanism of ER stress/UPR-related VSMC death triggered by progerin.”

The authors screened the same set of genes in immortalized human HGPS cells (Fig. 2 D). They state on Page 4: "We next used quantitative real-time PCR to analyze immortalized HGPS patient-derived cells." I simply want to check whether indeed they used immortalized or primary fibroblasts lines. If immortalized cells were used please comment of this choice?

RESPONSE: hTERT-immortalized HGPS patient-derived cells were used in Fig. 2D due to limited availability and reduced replicative potential of primary patient-derived fibroblasts. This cell line has been widely used (Fernandez et al, 2014; Scaffidi & Misteli, 2011), and we have confirmed that the immortalization process does not affect or contribute to the progerin-induced phenotypes.

Next, it has now been established that the level of progerin expression in HGPS cells is critical for inducing cellular malfunctions and most likely for ER stress. So, it would be important to show a progerin or lamin A/C western blot detection of HGPS cell extracts used in this study. Hence, progerin expression levels is variable in different tissues in HGPS mice and might as well influence the difference observed in gene expression profiles.

RESPONSE: According to the Reviewer's suggestion, we have performed western blots to examine the expression of lamin A, lamin C and progerin in various organs (medial aorta, heart, spleen, kidney, and liver) from mice used in this study (**Figure RV9**), as well as in immortalized HGPS patient-derived fibroblasts (**Figure RV10**). Previously, it was demonstrated in an elegant way that lamin A expression differs between organs and correlates with tissue stiffness (Swift & Discher, 2014; Swift et al, 2013). Here, we have confirmed that aortic media has a very high expression of lamin A/C (and thus progerin), indicating that indeed ER stress activation in this tissue might be related to higher expression of the mutant protein. To highlight the importance of this issue, we have included the following text in the Discussion section (page 6, first paragraph): "ER stress activation in other tissues of progeroid mice was variable. This variability might be at least in part attributed to different levels of lamin A (and thus progerin) expression associated with differences in tissue stiffness, with lower expression in softer tissues (Swift & Discher, 2014; Swift et al, 2013). Moreover, distinct organs and tissues may have different thresholds for tolerating misfolded protein load and therefore be more prone or resistant to ER stress."

3) The immunocytochemistry staining (Fig. EV3) support the conclusion that there is an increase ER stress as indicated in aortas sections from mice models expressing progerin. It is a pity that there is no mouse specific anti-progerin antibody available to visualize this effector over these markers.

RESPONSE: As pointed out by the Reviewer, there is no commercially-available antibody specifically recognizing mouse progerin. There used to be a polyclonal antibody from Santa Cruz Biotechnologies (discontinued product) made against the C-terminal part of prelamin A, which was also recognizing progerin and lamin A, but not lamin C. This antibody is not suitable for analyzing progerin expression in the ubiquitous progeria model, but we used it for progerin detection by immunofluorescence in *Lmna^{LCS/LCS}* animals that express only lamin C (and progerin upon Cre expression). However, since this antibody is made in rabbit, as all the antibodies we used for the ER stress detection (BiP, PDI, calreticulin), double staining is not possible. Nonetheless, the expression of progerin in VSMCs using this antibody was shown in our previous work (Hamczyk et al, 2018), as well as in **Figure RV5** (artery in the kidney) and **Figure RV1** (aorta in the embryo at 15.5 dpc).

4) To further assess the implication of ER stress response and UPR signaling in VSMC death and atherosclerosis in the ubiquitous and VSMC- specific progeria mice. The authors tested TUDCA, a chemical chaperone. The Chromatin and SMA staining of aorta sections convincingly support their conclusion that TUDCA treatment alleviates aortic VSMC loss in both mice models (Fig.3). Consequently, TUDCA treatment seems to prevent atherosclerosis development.

There is one sticking observation for the significant amelioration of the vascular system; the lifespan of Apoe^{-/-}-LmnaG609G/G609G mice is not increased. The authors refer to their previous publication (Hamczyk et al, 2018b) indicating that they die from different causes than vascular disease. A summary of the reasons causing their early death, independent of vascular disease will be helpful to be included in this publication. This information might be critical in vascular disease of individuals suffering from other syndromes than HGPS.

RESPONSE: We have included this information in the Discussion section (page 7, first paragraph): "This is in accordance with previous observations that, unlike HGPS patients, these mutant mice apparently die from atherosclerosis-independent causes, possibly arrhythmias, starvation and cachexia (Hamczyk et al, 2018; Kreienkamp et al, 2018)."

Finally, this study reports very fascinating and important findings on the potential link between progerin-ER stress and vascular disease, the most critical pathology of HGPS children. I am sure this study will attract readers from various fields beside HGPS.

RESPONSE: We are very grateful to the Reviewer for his/her positive remarks and enthusiasm for our study.

Referee #3 (Remarks for Author):

The same group has recently reported the generation of mouse models exhibiting progerin-induced acceleration of atherosclerosis, similarly to HFPS patients, identifying VSMCs as a major target for progerin in the vessel wall (Circulation 2018). Here, the authors performed RNA seq of the tunica media from APOE-KO mouse aorta with ubiquitous and VSMC-specific progerin expression with the aim of identifying progeria-induced VSMC-death and accelerated atherosclerosis. Results of high-throughput sequencing indicated that ER stress response and associated UPR gene expression are significantly affected in progerin-expressing VSMCs. Similar, these pathways were found altered in fibroblasts of HGPS patients. The authors used TUDCA for alleviating ER stress/UPR in their models, demonstrating a delayed VSMC loss and reduced atherosclerosis, importantly with life-span prolongation.

The study is well conducted. I have only some minor considerations:

a. Quantification of immunostaining in Figure EV3 should be provided. I suggest also to move the figure to the main manuscript as it includes relevant data.

RESPONSE: As requested, a quantification of immunostaining in **Figure EV3** is provided in the revised manuscript. To enable robust statistical analysis, we have analyzed 4 additional animals per genotype (total of 6-9 animals per genotype were tested). Moreover, as suggested by the Reviewer, the figure has been moved to the main manuscript (now **Figure 3**).

b. The same authors in a recent Circulation manuscript demonstrated that atherosclerosis increase in mice expression progerin is not associated with increased serum cholesterol levels. Could this point be commented? Does it have implication for data interpretation of this manuscript?

RESPONSE: Indeed, elevated cholesterol level is linked to increased atherosclerosis and can induce ER stress, thus the results demonstrating that progerin-expressing mice (both ubiquitous and VSMC-specific) show serum cholesterol concentrations similar to control mice was crucial for data interpretation. Namely, elevated serum cholesterol level is not responsible for either the ER stress activation in VSMC or the augmented atherosclerosis burden in progeroid mice. This subject is briefly commented in the Discussion section in the revised manuscript (page 6, first paragraph): “Given the importance of VSMCs in progerin-driven cardiovascular disease (Hamczyk et al, 2018; Olive et al, 2010; Stehbens et al, 2001), we sought to identify mechanisms underlying VSMC death and enhanced atherosclerosis in HGPS, which seem to be independent of elevated cholesterol levels in the blood (Gordon et al, 2005; Hamczyk et al, 2018).”.

c. For helping reading to an audience not familiar with progeria mouse models, I suggest to include in the M&M section a brief description of the *Lmna* L^{CSL}/L^{CS} mouse model used as control in some experiments.

RESPONSE: Following the suggestion of the Reviewer, a description of *Lmna*^{L^{CSL}/L^{CS}} mouse model has been included in the Materials and Methods section (page 8, second paragraph): “The *Lmna*^{L^{CS}} (Lamin C-STOP) allele consists of a modified *Lmna* gene in which a neomycin resistance gene flanked with two *loxP* sequences was introduced after exon 10 to abolish lamin A production. Additionally, an HGPS-equivalent point mutation (c.1827C>T; p.G609G) was inserted in exon 11. Excision of the neomycin resistance cassette in the presence of Cre recombinase enables progerin production (as well as lamin C and some residual lamin A). *ApoE*^{-/-}*Lmna*^{L^{CSL}/L^{CS}} mice show slightly increased weight and survival compared to wild-type controls (Lopez-Mejia et al, 2014).”

Figure RV1. Progerin expression in the aorta of *Apoe^{-/-}Lmna^{LCS/LCS}SM22aCre* mice during embryonic development. Embryos were collected at 11.5 and 15.5 days post coitum (dpc), embedded in paraffin, sectioned longitudinally and stained with Hoechst 33342 nucleic acid stain (visualized in blue), anti-smooth muscle α -actin (SMA, visualized in red) and anti-progerin (visualized in white) antibodies. Confocal microscopy images show aorta at 11.5 (A) and 15.5 dpc (B).

Figure RV2. Progerin expression in the heart of *Apoe^{-/-}Lmna^{LCS/LCS}SM22αCre* mice during embryonic development. Embryos were collected at 11.5 and 15.5 days post coitum (dpc), embedded in paraffin, sectioned longitudinally and stained with Hoechst 33342 nucleic acid stain (visualized in blue), anti-smooth muscle α -actin (SMA, visualized in red) and anti-progerin (visualized in white) antibodies. Confocal microscopy images show heart at 11.5 (**A**) and 15.5 dpc (**B**). Yellow arrows indicate examples of progerin-positive nuclei.

Figure RV3. Progerin expression in the intestine of *Apoe^{-/-} Lmna^{LCS/LCS} SM22αCre* mice during embryonic development. Embryos were collected at 11.5 and 15.5 days post coitum (dpc), embedded in paraffin, sectioned longitudinally and stained with Hoechst 33342 nucleic acid stain (visualized in blue), anti-smooth muscle α -actin (SMA, visualized in red) and anti-progerin (visualized in white) antibodies. Confocal microscopy images show intestine at 11.5 (A) and 15.5 dpc (B). Yellow arrows indicate examples of progerin-positive nuclei.

Figure RV4. Progerin expression in the kidney of *Apoe^{-/-}Lmna^{LCS/LCS}SM22αCre* mice during embryonic development. Embryos were collected at 11.5 and 15.5 days post coitum (dpc), embedded in paraffin, sectioned longitudinally and stained with Hoechst 33342 nucleic acid stain (visualized in blue), anti-smooth muscle α -actin (SMA, visualized in red) and anti-progerin (visualized in white) antibodies. Confocal microscopy images show kidney at 11.5 (A) and 15.5 dpc (B).

Figure RV5. Kidneys of 26-week-old *Apoe^{-/-}Lmna^{LCS/LCS}SM22αCre* mice have very low expression of progerin, predominantly limited to blood vessels. Male mice were fed normal chow and sacrificed at 26 weeks of age. Kidneys were embedded in paraffin and sectioned for immunofluorescence studies. Images show representative sections of the kidney (cortex, medulla and artery) stained against smooth muscle α -actin (SMA, visualized in red) and progerin (visualized in white). Nuclei were stained with Hoechst 33342 (visualized in blue). 4-6 animals per genotype were analyzed. Scale bar 50 μ m. Yellow arrows indicate examples of progerin-positive nuclei.

Figure RV6. Progerin expression in the muscularis externa in the intestine of 12- and 26-week-old *Apoe*^{-/-} *Lmna*^{LCS/LCS} *SM22αCre* mice does not cause smooth muscle cell loss. Male mice were fed normal chow and sacrificed at 12 or 26 weeks of age. Various regions of the intestine (duodenum, jejunum, ileum, and colon) were embedded in the paraffin and sectioned for immunofluorescence studies. (**A**, **C**) Images show transversal sections of the intestine stained against smooth muscle α -actin (SMA, visualized in red) and progerin (visualized in white) in 12- (**A**) and 26-week-old (**C**) animals. Nuclei were stained with Hoechst 33342 (visualized in blue). Scale bar 50 μ m. me – muscularis externa; mc – mucosa. Yellow arrows

indicate examples of progerin-positive nuclei. (B, D) Graphs represent quantification of the percentage of SMA-positive area in the muscularis externa in both circular and longitudinal muscle of 12- (B) and 26-week-old (D) animals. n=4-6. Data are presented as mean \pm SEM. Statistical differences were analyzed by two-tailed *t*-test.

Figure RV7. Intestine of *Apoe*^{-/-}*Lmna*^{LCS/LCS}*SM22aCre* mice does not present any evident pathological changes at 12 and 26 weeks of age. Male mice were fed normal chow and sacrificed at 12 or 26 weeks of age. Various regions of the intestine were embedded in paraffin. Images show hematoxylin&eosin-stained sections of the small intestine (duodenum, jejunum, and ileum) and large intestine (colon). 4-6 animals per condition were analyzed. Scale bar 300 μ m.

C

Gene symbol	Ubiquitous progerin				VSMC-specific progerin			
	RNAseq		qPCR		RNAseq		qPCR	
	Fold change	adjusted P value	Fold change	P value	Fold change	adjusted P value	Fold change	P value
Cse1l	1.03	0.883	1.12	0.062	1.26	0.025	1.19	0.001
Kpna1	1.11	0.501	1.17	0.031	1.29	0.022	1.27	0.003
Kpna3	1.05	0.813	1.31	0.002	1.31	0.023	1.14	0.017
Kpna4	0.93	0.717	1.05	0.315	1.32	0.020	1.30	0.012
Ran	0.95	0.763	1.06	0.153	1.27	0.009	1.06	0.077
Rcc1	0.98	0.967	1.08	0.196	1.45	0.043	1.36	0.008
Xpo1	1.41	0.001	1.28	0.014	1.34	0.008	1.15	0.175
Rangap1	0.95	0.815	0.92	0.220	1.29	0.022	1.29	0.030
Tnp1	1.09	0.613	0.99	0.386	1.22	0.109	1.01	0.496

Figure RV8. RAN signaling in medial aortas from *Apoe*^{-/-}*Lmna*^{G609G/G609G} and *Apoe*^{-/-}*Lmna*^{LCS/LCS} SM22aCre mice. Male mice fed normal chow were sacrificed at 8 weeks of age. Aortas were extracted, cleaned of fatty tissue and digested with collagenase II to obtain aortic media. Medial aortas from 3-5 animals of the same genotype were pooled. RNA was extracted, retrotranscribed to cDNA and real-time qPCR was performed using SYBRGreen. qPCR primers used for RAN signaling pathway validation are shown in the table below. (A, B) Graphs show mRNA levels (as fold change, FC) of the above-mentioned genes in medial aortas obtained from *Apoe*^{-/-}*Lmna*^{G609G/G609G} mice (A) and *Apoe*^{-/-}*Lmna*^{LCS/LCS} SM22aCre mice (B) compared to their corresponding controls (*Apoe*^{-/-}*Lmna*^{+/+} and *Apoe*^{-/-}*Lmna*^{LCS/LCS} mice, respectively). n=3-4 pooled samples per genotype, each pool proceeded from 3-5 animals. *Hprt* and *Gusb* were used for normalization. *Tnp1* was used as a negative control (gene within RAN signaling pathway that was not found upregulated in the *Apoe*^{-/-}*Lmna*^{LCS/LCS} SM22aCre model in the RNAseq study). Data are presented as mean ± SEM. Statistical differences were analyzed by one-tailed *t*-test. **P*<0.05, ***P*<0.01, ****P*<0.001. (C) Table summarizing RNAseq and qPCR results for the selected genes within RAN pathway. Cells highlighted in gray indicate results that reached statistical significance (*P*<0.05).

Gene name	Forward primer sequence (5' - 3')	Reverse primer sequence (5' - 3')
Cse1l	TCAGAATTACCCACTGTTGCT	GCTACTCGATCAGCTTCGCA
Kpna1	ATTCTCGAATAGCCCAGAGC	TCCACAAACCTGGCCACTAC
Kpna3	TCCAGTGACAGAAATCCACCA	GTCTGTGCAGAAGTTCCCGA
Kpna4	TCCAGTGATCGAAATCCACCA	TGCTTGCGTTTTGTTTCAGACG
Ran	AGACAGGAAAGTGAAGGCAA	TGGCAAGCCAGAGGAAAGG
Rangap1	GCCATGGCCTCTGAAGACAT	CTTCGAGGCCGTCAAACCTCT
Rcc1	GGGCTTCTGTGGTTATGCT	TCCAGCTGTTTGCCAGTCAT
Tnp1	AGCTGCAGAATTGGCCTGAT	TGGGACTGCTGTGCTTGAAA
Xpo1	TCATTTGGCAGCTGAGCTCT	AGCCATGCGACTAACCATCA

Figure RV9. Progerin expression in different organs from *Apoe*^{-/-}*Lmna*^{G609G/G609G} and *Apoe*^{-/-}*Lmna*^{LCS/LCS}*SM22aCre* mice. Male mice were fed normal chow and sacrificed at 8 weeks of age. Heart, spleen, kidney, liver and aorta were extracted. Aortas were digested with collagenase II to obtain medial tissue containing vascular smooth muscle cells. Due to low amount of the sample, medial aortas from approx. 5 animals of the same genotype were pooled. Frozen tissues (except for medial aorta) were pulverized with BioPulverizer (BioSpec Products). All samples were then homogenized using TissueLyser (Qiagen) and urea buffer containing 50 mM Tris-HCl pH 8.8, 2% sodium dodecyl sulfate (SDS), 8 M urea, 2 M thiourea, and inhibitors of proteases and phosphatases. Tissue lysates were sonicated using Bioruptor (Diagenode), incubated for 20 minutes at 4°C (with mixing), and centrifuged to pellet insoluble material. Protein concentration in the supernatant was measured using Bradford assay. Samples were loaded on 4–20% Criterion TGX precast gels (26 well, 5671095, BioRad). For heart, spleen, kidney and liver, 14 µg of total protein lysate were loaded on each well. For medial aorta, 3.5 and 7 µg of total protein lysate were loaded on each well. Following electrophoresis, proteins were transferred to polyvinylidene fluoride (PVDF) membranes, which were blocked with 5% bovine serum albumin (BSA, A7906, Sigma-Aldrich) in Tris-buffered saline (TBS) containing 0.02% Tween20 (P1379, Sigma-Aldrich). After blocking, membranes were incubated overnight at 4°C with primary antibodies diluted in 5% BSA, 0.02% Tween20 in TBS. A-type lamins, including lamin A, lamin C and progerin, were detected using anti-lamin A/C antibody (1:500, sc-376248, Santa Cruz Biotechnology). Loading controls were GAPDH (1:15000, MAB374, Millipore) and vinculin (1:1000, V9131, Sigma-Aldrich) (metavinculin is an isoform present in smooth muscle and skeletal muscle). Membranes were incubated for 1 hour at room temperature with horseradish peroxidase (HRP)-conjugated secondary antibody (1:2500-1:15000, sc-516102, Santa Cruz Biotechnology) diluted in 5% BSA, 0.02% Tween20 in TBS. Luminata Classico Western HRP Substrate (WBLUC0100, Millipore) was used for detection. Chemiluminescence images were acquired using ImageQuant LAS 4000 mini system (GE Healthcare Life Sciences). **(A)** Western blot analysis of lamin A/C and progerin expression in *Apoe*^{-/-}*Lmna*^{G609G/G609G} mice (with ubiquitous progerin expression) and in control *Apoe*^{-/-}*Lmna*^{+/+} mice (expressing both lamin A and C). **(B)** Western blot analysis of lamin A/C and progerin expression in *Apoe*^{-/-}*Lmna*^{LCS/LCS}*SM22aCre* mice (with vascular smooth muscle cell-specific progerin expression) and in control *Apoe*^{-/-}*Lmna*^{LCS/LCS} mice (only expressing lamin C).

Figure RV10. Progerin expression in patient-derived HGPS fibroblasts. Total cell lysates were prepared by incubating 7.5×10^5 cells with $150 \mu\text{l}$ of $1 \times$ SDS sample buffer (Bio-Rad). Lysates were separated by SDS-polyacrylamide gel electrophoresis and transferred onto a polyvinylidene fluoride (PVDF) membrane, which was blocked for 30 minutes in blocking buffer (5% BSA in TBS containing 0.05% Tween 20). After blocking, the membrane was incubated overnight at 4°C with primary antibodies diluted in 2% BSA/TBS containing 0.05% Tween 20. Primary antibodies used were mouse monoclonal anti-lamin A/C antibody (1:5000, clone E1, sc-376248, Santa Cruz Biotechnology) and mouse monoclonal anti-GAPDH antibody (1:20000, clone 6C5, ab8245, Abcam), which served as a loading control. Membranes were washed three times for 15 minutes in 0.05% Tween 20-TBS and incubated with horseradish peroxidase-labeled secondary antibodies (Santa Cruz Biotechnologies) for 1 hour at room temperature. After washing, chemiluminescence images were acquired using ChemiDoc system (BioRad). Figure shows Western blot analysis of total lamin A/C and progerin levels in patient-derived immortalized human HGPS fibroblasts and healthy control.

References

- de la Rosa J, Freije JM, Cabanillas R, Osorio FG, Fraga MF, Fernandez-Garcia MS, Rad R, Fanjul V, Ugalde AP, Liang Q et al (2013) Prelamin A causes progeria through cell-extrinsic mechanisms and prevents cancer invasion. *Nat Commun* 4: 2268
- Fernandez P, Scaffidi P, Markert E, Lee JH, Rane S, Misteli T (2014) Transformation resistance in a premature aging disorder identifies a tumor-protective function of BRD4. *Cell Rep* 9: 248-260
- Gordon LB, Harten IA, Patti ME, Lichtenstein AH (2005) Reduced adiponectin and HDL cholesterol without elevated C-reactive protein: clues to the biology of premature atherosclerosis in Hutchinson-Gilford Progeria Syndrome. *J Pediatr* 146: 336-341
- Hamczyk MR, Villa-Bellosta R, Gonzalo P, Andres-Manzano MJ, Nogales P, Bentzon JF, Lopez-Otin C, Andres V (2018) Vascular Smooth Muscle-Specific Progerin Expression Accelerates Atherosclerosis and Death in a Mouse Model of Hutchinson-Gilford Progeria Syndrome. *Circulation* 138: 266-282
- Hennekam RC (2006) Hutchinson-Gilford progeria syndrome: review of the phenotype. *Am J Med Genet A* 140: 2603-2624
- Kreienkamp R, Billon C, Bedia-Diaz G, Albert CJ, Toth Z, Butler AA, McBride-Gagyi S, Ford DA, Baldan A, Burris TP et al (2018) Doubled lifespan and patient-like pathologies in progeria mice fed high-fat diet. *Aging Cell*: e12852
- Lopez-Mejia IC, de Toledo M, Chavey C, Lapasset L, Cavelier P, Lopez-Herrera C, Chebli K, Fort P, Beranger G, Fajas L et al (2014) Antagonistic functions of LMNA isoforms in energy expenditure and lifespan. *EMBO Rep* 15: 529-539
- Merideth MA, Gordon LB, Clauss S, Sachdev V, Smith AC, Perry MB, Brewer CC, Zalewski C, Kim HJ, Solomon B et al (2008) Phenotype and course of Hutchinson-Gilford progeria syndrome. *N Engl J Med* 358: 592-604
- Olive M, Harten I, Mitchell R, Beers JK, Djabali K, Cao K, Erdos MR, Blair C, Funke B, Smoot L et al (2010) Cardiovascular pathology in Hutchinson-Gilford progeria: correlation with the vascular pathology of aging. *Arterioscler Thromb Vasc Biol* 30: 2301-2309
- Prather RS, Kubiak J, Maul GG, First NL, Schatten G (1991) The expression of nuclear lamin A and C epitopes is regulated by the developmental stage of the cytoplasm in mouse oocytes or embryos. *J Exp Zool* 257: 110-114
- Rober RA, Weber K, Osborn M (1989) Differential timing of nuclear lamin A/C expression in the various organs of the mouse embryo and the young animal: a developmental study. *Development* 105: 365-378
- Scaffidi P, Misteli T (2011) In vitro generation of human cells with cancer stem cell properties. *Nat Cell Biol* 13: 1051-1061
- Schatten G, Maul GG, Schatten H, Chaly N, Simerly C, Balczon R, Brown DL (1985) Nuclear lamins and peripheral nuclear antigens during fertilization and embryogenesis in mice and sea urchins. *Proc Natl Acad Sci U S A* 82: 4727-4731
- Stehbens WE, Delahunt B, Shozawa T, Gilbert-Barnes E (2001) Smooth muscle cell depletion and collagen types in progeric arteries. *Cardiovasc Pathol* 10: 133-136

Stewart C, Burke B (1987) Teratocarcinoma stem cells and early mouse embryos contain only a single major lamin polypeptide closely resembling lamin B. *Cell* 51: 383-392

Sullivan T, Escalante-Alcalde D, Bhatt H, Anver M, Bhat N, Nagashima K, Stewart CL, Burke B (1999) Loss of A-type lamin expression compromises nuclear envelope integrity leading to muscular dystrophy. *J Cell Biol* 147: 913-920

Swift J, Discher DE (2014) The nuclear lamina is mechano-responsive to ECM elasticity in mature tissue. *J Cell Sci* 127: 3005-3015

Swift J, Ivanovska IL, Buxboim A, Harada T, Dingal PC, Pinter J, Pajerowski JD, Spinler KR, Shin JW, Tewari M et al (2013) Nuclear lamin-A scales with tissue stiffness and enhances matrix-directed differentiation. *Science* 341: 1240104

Uppala JK, Gani AR, Ramaiah KVA (2017) Chemical chaperone, TUDCA unlike PBA, mitigates protein aggregation efficiently and resists ER and non-ER stress induced HepG2 cell death. *Sci Rep* 7: 3831

Vandewynckel YP, Laukens D, Devisscher L, Paridaens A, Bogaerts E, Verhelst X, Van den Bussche A, Raevens S, Van Steenkiste C, Van Troys M et al (2015) Tauroursodeoxycholic acid dampens oncogenic apoptosis induced by endoplasmic reticulum stress during hepatocarcinogen exposure. *Oncotarget* 6: 28011-28025

Thank you for the submission of your revised manuscript to EMBO Molecular Medicine. We have now received the enclosed reports from the three referees, who are supportive of publication. I am thus pleased to inform you that we will be able to accept your manuscript pending minor editorial amendments and a response to referee #1.

***** Reviewer's comments *****

Referee #1 (Remarks for Author):

The version 2 of the manuscript by Hamczyk et al provides answer to some of my questions, some of them resulting in other comments.

1. The evidence that TUDCA corrected UPR/ER stress pathway in mice model.
The figure EV3 added in the manuscript suggested that TUDCA does not correct ER stress. Even if Ddit3 mRNA decreased, mRNAs of two genes Pdia4 and Hsp90b1 increased in Apoe^{-/-} Lmna^{LCS/LCSSM22αCre} mice, whereas three others (Calr, DNAjb9, HspA5) also increased but without statistical significance.
Moreover TUDCA does not modify the expression of the 6 ER-stress related genes in Apoe^{-/-} Lmna^{G609G/G609G} mice, thus clearly limiting the interest of the drug for the treatment of progeria patients, where progerin expression is ubiquitous.
These data require a specific detailed comment that is lacking both in the results and in the discussion section.

2. The control that progerin does not result in smooth muscle cell loss in other organs than vascular wall, either during embryonic development or in adult.
The results presented by the authors are clearly convincing. All the related figures could be added as supplemental data, including the related "material and methods" paragraph.

3. The manuscript does not explore the pathophysiological mechanism(s) resulting in VSMC loss. I agree with the author comment stating that this point requires further experiments.

4. Regarding HFD, the author answer pointed out that in figure 4, data from panels A to F were from HFD-fed mice, whereas panel G (TUDCA effect on mice survival) described mice receiving normal diet.
Survival data in HFD-fed mice receiving or not TUDCA are missing.
The summary has to be more accurate regarding HFD- and normal diet-fed mice, as well as the results section (bottom page 5).

As asked by referee 2, the authors presented in figure RV8 data regarding RAN signaling pathway. Does TUDCA correct the overexpression of RAN signaling pathway ?

Minor comment : page 7 in the discussion section : to add "heart valves defects" in the list of HGPS disorder, in line with histological data showing progerin expressing cells in cardiac valves.

Despite the interest of the topic, the quality of some of the reported data, the version 2 of the manuscript has to be improved to be published in EMBO Journal Mol Med.

Referee #2 (Remarks for Author):

I am satisfied with the corrections

Referee #3 (Remarks for Author):

No further question.

Referee #1 (Remarks for Author): The version 2 of the manuscript by Hamczyk et al provides answer to some of my questions, some of them resulting in other comments. 1. The evidence that TUDCA corrected UPR/ER stress pathway in mice model. The figure EV3 added in the manuscript suggested that TUDCA does not corrected ER stress. Even if *Ddit3* mRNA decreased, mRNAs of two genes *Pdia4* and *Hsp90b1* increased in *Apoe*^{-/-} *Lmna*^{LCS/LCSSM22α} *Cre* mice, whereas three others (*Calr*, *DNAjb9*, *HspA5*) also increased but without statistical significance. Moreover TUDCA does not modify the expression of the 6 ER-stress related genes in *Apoe*^{-/-} *Lmna*^{G609G/G609G} mice, thus clearly limiting the interest of the drug for the treatment of progeria patients, where progerin expression is ubiquitous. These data require a specific detailed comment that is lacking both in the results and in the discussion section.

RESPONSE: Please note that the expression of the six ER stress-related genes was not tested in TUDCA-treated *Apoe*^{-/-} *Lmna*^{G609G/G609G} mice. The left two bars in Fig EV3 indicate *Apoe*^{-/-} *Lmna*^{LCS/LCS} control mice injected with PBS or TUDCA. As expected, TUDCA did not elicit any effect in the control medial aortas that do not exhibit ER stress, but it reduced the *Ddit3* expression and increased the expression of *Pdia4* and *Hsp90b1* in progerin-expressing medial aortas, indicating modulation of ER stress/UPR response by TUDCA. It is important to highlight that TUDCA is not an inhibitor of ER stress in the strict meaning of the term inhibitor. Most chemical chaperones are thought to reduce the misfolded protein overload either directly by interacting with the misfolded proteins and chaperones or indirectly by inducing expression of genes involved in protein folding and expansion of the ER (Park SW and Ozcan U. *Semin Immunopathol* 35: 351-373, 2013). Thus, the increase in the expression of chaperones, co-chaperones and isomerases is to be expected in TUDCA-treated samples. If the ER stressor is temporary (e.g. short-term tunicamycin treatment), cells can first increase gene expression of the protein folding machinery leading to ER stress resolution and UPR shut down, and finally restore normal expression levels of ER-stress related genes. In progeria, however, progerin is being produced continuously, thus the only reasonable approach (other than to lower/block progerin expression) is to maintain the cells in a pro-survival phase of the ER stress since it cannot be resolved. Hence, a decrease in the expression of a pro-apoptotic UPR transcription factor, *Ddit3* is a desired beneficial effect of TUDCA. Some of the chemical chaperones can inhibit the ER stress response leading to cell death (Uppala JK. et al. *Sci Rep* 7: 3831, 2017), which can be beneficial in cancer; however, promoting cell death would be detrimental in progeria since it could further accelerate VSMC loss. Targeting ER stress response in disease consist in modulating it rather than correcting it as pointed out by C. Hetz, E. Chevet and H.P. Harding in their *Nature Reviews Drug Discovery* paper: "...the activation of UPR signalling can engage both pro-survival and pro-apoptotic cellular programmes. Thus, modulating UPR signalling components has the potential to either stimulate an increased capacity to alleviate protein misfolding, which could have therapeutic effects in PMDs, or to promote apoptosis, which could be used as an anticancer strategy." (Hetz C et al. *Nat Rev Drug Discov* 12:703-719, 2013). As suggested by the Referee, the results of Fig. EV3 are commented in more detail in the revised manuscript, both in the Results section (page 5: "One-week TUDCA treatment of 8-week-old *Apoe*^{-/-} *Lmna*^{LCS/LCSSM22α} *Cre* mice slightly increased in medial aorta the mRNA levels of *Pdia4* (coding for a protein disulfide isomerase) and *Hsp90b1* (coding for a chaperone), suggesting mild enhancement of protein folding capacity (Fig EV3). Moreover, it markedly decreased the expression of *Ddit3* gene coding for DDIT3, a pro-apoptotic transcription factor of the UPR machinery (Fig EV3), indicating that TUDCA helps to resist ER stress-induced death in VSMCs."), and the Discussion section (page 7: "At the molecular level, one-week *in vivo* TUDCA treatment caused a slight increase in the expression of some genes related to protein folding and modification in progerin-expressing medial aortas. Importantly, TUDCA diminished the mRNA levels of pro-apoptotic *Ddit3*, consistent with the known cytoprotective properties of this compound (Gavin et al, 2016; Rivard et al, 2007; Uppala et al, 2017; Xie et al, 2002). Overall, these results suggest that TUDCA simulates the pro-survival UPR and attenuates the pro-apoptotic UPR; however, the exact molecular mechanism of action of TUDCA in progerin-expressing cells remains to be explored in further details.").

2. The control that progerin does not result in smooth muscle cell loss in other organs than vascular wall, either during embryonic development or in adult. The results presented by the authors are clearly convincing. All the related figures could be added as supplemental data,

including the related "material and methods" paragraph.

RESPONSE: As suggested, we have included figures RV1-6 as supplemental information (Appendix Figures S3-8). Description of the corresponding methods was added to the Materials and Methods section.

3.The manuscript does not explore the pathophysiological mechanism(s) resulting in VSMC loss. I agree with the author comment stating that this point requires further experiments.

RESPONSE: We thank the Referee for his/her understanding.

4. Regarding HFD, the author answer pointed out that in figure 4, data from panels A to F were from HFD-fed mice, whereas panel G (TUDCA effect on mice survival) described mice receiving normal diet. Survival data in HFD-fed mice receiving or not TUDCA are missing. The summary has to be more accurate regarding HFD- and normal diet-fed mice, as well as the results section (bottom page 5).

RESPONSE: We have added extra information in “The paper explained”, and in the Discussion to more accurately indicate the use of normal diet and HFD (no extra information was added in the Abstract due to space constraints: 175-word limit). Furthermore, we have included in the last paragraph of the Results section sentences clearly introducing the experiments with HFD and normal chow diet to avoid any misinterpretation: “After confirming that prolonged TUDCA administration did not trigger any deleterious side effects (**Appendix Fig S2**), we evaluated its effectiveness in ameliorating vascular disease in high-fat diet-fed ubiquitous and VSMC-specific progeroid mouse models. TUDCA treatment alleviated aortic VSMC loss (**Fig 4A and B**), adventitial thickening (**Fig 4C and D**), and inhibited atherosclerosis (**Fig 4E and F**) in both *Apoe*^{-/-}*Lmna**G609G/G609G* and *Apoe*^{-/-}*Lmna**LCS/LCSSM22aCre* mice. Atheromas of TUDCA-treated progeroid animals showed reduced necrotic core size and increased VSMC content (**Table EV2**), indicating an amelioration of the vulnerable-plaque phenotype reported previously in these mice (Hamczyk et al, 2018b). We next assessed the effect of a sustained TUDCA treatment on survival in normal chow-fed progeroid mice. TUDCA prolonged the median lifespan of *Apoe*^{-/-}*Lmna**LCS/LCSSM22aCre* mice by 38% (median survival: 64.15 weeks in TUDCA-treated *versus* 46.45 weeks in untreated mice), without significantly affecting the survival of *Apoe*^{-/-}*Lmna**G609G/G609G* mice (**Fig 4G**).”

We performed the survival studies in mice fed normal chow because long-term HFD feeding increases the incidence and severity of ulcerative dermatitis, a spontaneous condition in mice with a C57BL/6 background (Hampton AL et al. *J Am Assoc Lab Anim Sci* 51: 586-593, 2012; Hampton AL et al. *J Am Assoc Lab Anim Sci* 54: 487-496, 2015; Neuhaus B et al. *Exp Dermatol* 21: 712-713, 2012). Mice affected by severe ulcerative dermatitis require euthanasia due to ethical reasons resulting in a limited number of mice that die naturally or meet human end points other than ulcerative dermatitis. Moreover, life-long use of HFD leads to diabetes and decreases longevity itself (Baur JA et al. *Nature* 444: 337-342, 2006; Zhu et al. *Aging Cell*: e12883, 2019). Hence, to avoid the long-term adverse effects of HFD, survival experiments with and without TUDCA treatment were performed in mice fed normal chow.

As asked by referee 2, the authors presented in figure RV8 data regarding RAN signaling pathway. Does TUDCA corrected the overexpression of RAN signaling pathway ?

RESPONSE: Please note that the experiment of Figure RV8 was performed only to respond to Referee’s 2 comment, but was not included in the manuscript, which is focused on the role of ER stress response in progeria. We respectfully submit that investigating the possible role of RAN signaling pathway is beyond the scope of the present work.

Minor comment: page 7 in the discussion section : to add "heart valves defects" in the list of HGPS disorder, in line with histological data showing progerin expressing cells in cardiac valves.

RESPONSE: We believe that the Reviewer refers to the following sentence: “This is in accordance with previous observations that, unlike HGPS patients, these mutant mice apparently die from atherosclerosis-independent causes, possibly arrhythmias, starvation and cachexia (Hamczyk et al,

2018; Kreienkamp et al, 2018).” Please note that “arrhythmias, starvation and cachexia” do not refer to general HGPS features, but only to those that have been suggested by previous studies as possible causes of death in the ubiquitous progeria mice. Even though valve defects might be connected with arrhythmias and were described in HGPS patients, we think that introducing “heart valves defects” in this particular sentence would not be accurate. However, to emphasize the presence of heart valve defects in some HGPS patients, we have modified a sentence in the Introduction section (page 3): “Progerin also provokes cardiac abnormalities (Merideth et al, 2008; Prakash et al, 2018; Rivera-Torres et al, 2016) and defects in heart valves (Hanumanthappa et al, 2011; Merideth et al, 2008; Nair et al, 2004; Olive et al, 2010) and blood vessels, including vascular smooth muscle cell (VSMC) loss, adventitial thickening, calcification, and extracellular matrix deposition (Olive et al, 2010; Stehbens et al, 2001; Stehbens et al, 1999).”

Despite the interest of the topic, the quality of some of the reported data, the version 2 of the manuscript have to be improved to be published in EMBO Journal Mol Med.

RESPONSE: We thank the Referee for making additional suggestions to improve our work.